# Stabilising large grains in self-forming, steep channels

William H. Booker[1] and Brett C. Eaton[1]

[1]Geography Department, University of British Columbia, Vancouver, BC, Canada

**Correspondence:** William H. Booker (william.booker@alumni.ubc.ca)

**Abstract.** It is understood that the interaction between sediment supply and discharge drives first-order behaviour of alluvial deposits. The influence of the grain size distribution over the mobility and resultant evolution is, however, unclear. Four experiments were conducted in a scaled physical model for two grain size distributions, analogous to a one-dimensional self-formed alluvial fan. We demonstrate the unsuitability of the median grain size as a predictor of deposit behaviour at flows when the material is not equally mobile. The results instead suggest, during conditions of unequal mobility, that the largest grains control the transport efficiency of the overall sediment mixture, and thus also the morphodynamics of the deposit and its tendency to store or evacuate material. Deposits appear to show a dependence upon the rate of material supply more strongly when the likelihood of its motion is less equally distributed (i.e., under partial transport conditions). If the coarse fraction (e.g., greater than $84^{th}$ percentile) is instead mobile due to increased discharge or because of their relative size, transport rates will increase and the behaviour of the mixtures converge to a common state, with morphology influenced by the material's mobility.

## 1 Introduction

Gravel bed rivers adjust their boundaries from the grain to the reach scale in response to the supplied sediment and water discharges (Leopold and Maddock, 1953; Lane, 1955; Howard and Kerby, 1983; Madej and Ozaki, 1996; Eaton and Church, 2004; Hassan et al., 2007). The feedbacks and interactions between antecedent flow and sediment discharge control river channel form (Fukuoka, 1989) and thereby influence channel response, for example as a response to flow increases (Masteller and Finnegan, 2017). Natural channels are likely to experience a distribution of flow rates and, therefore, corresponding modes of transport (e.g., Ashworth and Ferguson, 1989; Warburton, 1992). Central to the behaviour of gravel bed rivers is this response to their environment as the flow does not regularly or greatly exceed the threshold of sediment mobility (e.g., Church, 2006). In a channel the aggradation or degradation of material will lead to changes in its elevation, representing a balance between the amount of energy and material provided to it. Lane (1955) proposed that grade represents this balance as:

$$\frac{Q_b}{QS} \propto \frac{1}{D} \tag{1}$$

wherein the left hand side of the proportionality represents the sediment transport efficiency, given by the ratio between sediment supply ($Q_b$) and the product of discharge ($Q$) and slope ($S$). The right hand side is the reciprocal of the sediment flux calibre ($D$). Church (2006) recast this relation in a dimensionally balanced version, which can be written as follows:

$$\frac{Q_b}{QS} \propto \frac{d}{D} \tag{2}$$

where $d$ is the flow depth. In Church's version, $D$ is specifically defined to be the median bed surface size, based on the understanding of the hiding/exposure processes controlling the entrainment of sediment from a mixture.

Accordingly, mixtures of the same median grain size, under the same water and sediment discharges, should form to the same slope because of their equal transport efficiency. Transport efficiency ($\eta$) is defined in the same manner as by Bagnold (1966), in that it relates the work rate of the flow to the stream power available and describes the efficiency of the system in converting stream power into work (i.e., sediment transport) and is therefore higher in more efficient systems. That is, systems with higher $\eta$ values will organise to lower slopes because it is more capable of transporting the supplied material, as described by its discharge. Here, it is reformulated neglecting the mass flux term from its original form, instead replicating the dimensionless, volumetric consideration used by Eaton and Church (2011):

$$\eta = \frac{Q_b}{QS} \tag{3}$$

whereby it functions as a relationship between the system's mass and energy inputs, outputs and processes.

The validity of using a single characteristic grain size as a descriptor of a whole system's state is, however, fundamentally flawed. We know that surficial adjustment, bed forms and macroforms modulate bed material sediment transport rate, acting to dissipate energy and provide stability to the overall channel (Cherkauer, 1973; Montgomery and Buffington, 1997; Venditti et al., 2017). For example, it has been thought that grains may stabilise through rotation (Masteller and Finnegan, 2017), their organisation into cells (Church et al., 1998; Monsalve and Yager, 2017) and the formation of alternate bars and patches (Lisle et al., 1991; Dietrich et al., 2006; Nelson et al., 2010). One of the most well studied of these adjustment phenomena is the coarsening of the bed surface due to the preferential removal of fines or their kinetic sieving into the subsurface, until an armour layer develops that approximately equalises the threshold entrainment stress of the bed (Parker and Klingeman, 1982; Parker et al., 1982b; Andrews, 1983). Armour may develop in both sediment-starved reaches as static armour (Sutherland, 1987; Dietrich et al., 1989; Parker and Sutherland, 1990; Kondolf, 1997; Vericat et al., 2006; Nelson et al., 2009), or where sediment supply is present as a mobile armour layer (Parker and Klingeman, 1982; Andrews and Parker, 1987; Parker, 1990). It is the formation of an armour layer that prevents continued transport of the material and stabilises the channel against further deformative work. In addition, MacKenzie and Eaton (2017) demonstrated that it is the largest grains found in the bed material that control channel stability during degradation because of their role in protecting the underlying fine grains. Their work concludes that bed stability cannot be fundamentally linked to the median bed surface grain size, as in sediment transport models developed by Parker (1990) and Wilcock and Crowe (2003).

In contrast to this, our knowledge of the processes stabilising aggrading systems is substantially lacking in direct study; the omission of their explicit focus in the Treatise of Geomprohology is noticeable, in comparison to the myriad studies based in degrading channels. Systems undergoing aggradation may range in degrees of confinement (i.e., from valley bottoms to alluvial fans), but are characterised by either a competence or capacity driven accumulation of sediment relative to their current energy flux. Aggrading systems are often studied, but are often treated at a greater scale (i.e., channel planform) in the field (e.g., Gilbert, 1917; Harvey, 1991; Benda and Dunne, 1997) or neglected in non-fan experiments. For example, Madej (1982) attributed increases in sediment transport rates to channel geometry changes induced by aggradation in the channel, rather than

the manifestation of system variables such as slope (as would be expected with Eq. (1)). As a singular process, avulsion acts as a mechanism for channel 'stabilisation' in aggrading systems, where sediment transport capacity is maintained through the creation or re-occupation of an alternate channel position (Ashmore, 1982; Field, 2001). Studies also focus upon the influence of supplied material, of which the calibre is important for the resulting trajectory of changes to hydraulics and morphology.

An influx of fine sediment will increase sediment transport through increased exposure effects on coarser material (Wilcock et al., 2001; Wilcock and Crowe, 2003; Curran and Wilcock, 2005). On the other hand, coarse material will accumulate either through supply of unentrainable material (Harvey, 2001) or the role of coarse grains as stabilising loci (Lisle et al., 1991). We could argue, therefore, that there exists a precedent for the role of large grains in controlling the behaviour of aggrading channels, derived from the deposition of those grains supplied to the channel (e.g., Moss, 1963; Dunkerley, 1990).

The applicability of using Eq. (2) to predict the changes of system slope is thus called into question when we consider the role of large grains in the stability of aggrading or degrading systems. We hypothesise that the presence of the large grains will result in different sediment mixture mobilities for aggrading channels, thus different channel morphodynamics and depositional slope, as in MacKenzie and Eaton (2017). In addition, we expect that this effect will not be maintained under discharge increases, as the $D_{84}$ is suggested to strongly influence the thresholds of mobility within a mixture (MacKenzie

et al., 2018). The goal of this paper, therefore, is to test whether or not large grains influence channel stability in aggrading systems, wherein many of the processes thought to produce stabilisation in degrading systems are suppressed. To that end, we present the results for two sets of four experiments paired by median grain size but differentiated by the shape of their distributions, for which Eq. (1) and Eq. (2) would predict similar behaviour. In most studies, slope acts as a response of an existing deposit; for example, degradation into a bed surface (e.g., Parker et al., 1982a). Here, sediment may freely aggrade or

degrade and thus slope acts instead as an emergent indicator of the system state, thus allowing its form to fully represent the suite of process acting upon it, a methodology reserved mostly for fan studies (e.g., Schumm et al., 1987; Clarke et al., 2010). The results described here show that the grain size distribution used affects the resulting behaviour of the deposit and its slope, and the differences between paired experiments are controlled by the experimental boundary conditions.

## 2  Methods

Eight experiments were run in the recently constructed steep mountain channel flume at the University of British Columbia. The flume is acrylic walled, 2 m long by 0.128 m wide with a foam insert creating a transition from a steep (slope = 0.1 m/m) upper and flat (slope = 0 m/m) lower section (Figure 1), upon which a fan deposit can develop. These deposits that form within the flume are analogous to a one dimensional fan, or to the channel bed of a steep river confined by bedrock walls. Design and methodological cues were taken from previous experiments concerning self formed deposition (Guerit et al., 2014) and steep

channel stability (Lisle et al., 1991).

During the runs reported here, feed and flow were held constant for the length of each experiment. These were conducted under one of four conditions: 100L, 100H, 200L or 200H, where the number refers to the flow rate (in ml s$^{-1}$) and the letter to the feed rate (L = 1 g s$^{-1}$, H = 2 g s$^{-1}$). The experiments also have a relative sediment concentration compared to the

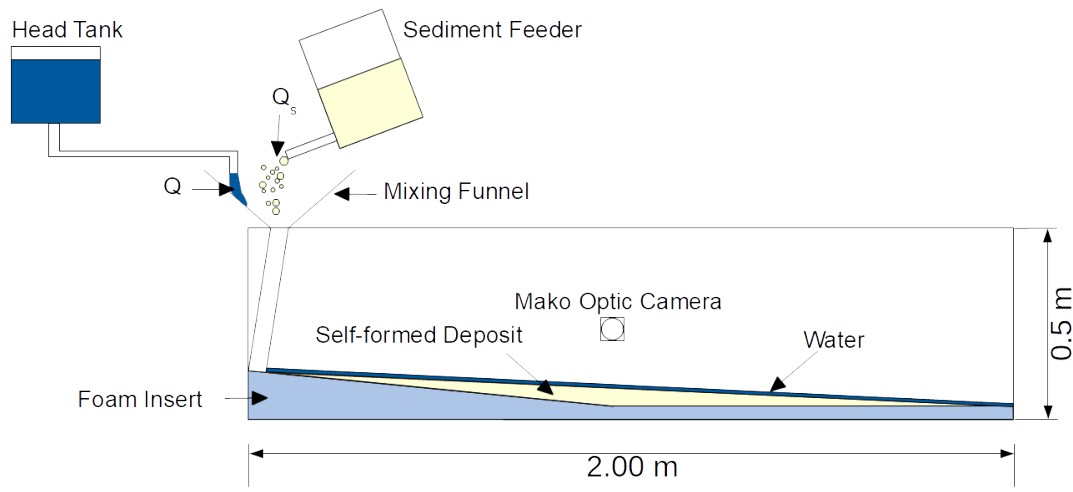

**Figure 1.** Simplified diagram of the experimental setup for the steep channel flume.

100L experiment, where a value of 1 represents both factors increasing (i.e., 100L and 200H), 0.5 is a halving of feed relative to flow (i.e., 200L) and 2 is a halving of flow relative to feed (i.e., 100H). This range of values allows us to consider five changes in relative sediment concentration mediated by changes to discharge or sediment feed rate. These are: (1) no change in concentration but changes in the total flux magnitude (100L vs. 200H), (2) doubling concentration through increasing sediment

feed (100L vs. 100H), (3) doubling concentration through decreasing discharge (200H vs. 100H), (4) halving concentration through increasing discharge (100L vs. 200L) and (5) halving concentration through sediment feed (200H vs. 200L). As in MacKenzie and Eaton (2017), sediment is scaled from gravel-bedded streams found in Alberta, Canada and truncated at 0.25 mm at the lower end to remove unscalable laminar sub-layer effects for sediment finer than this size limit (Peakall et al., 1996).

These eight experiments primarily serve to distinguish between the behaviour of two grain size distributions across a range

of run conditions. The two grain size distributions share nearly the same $D_{50}$ (GSD$_{broad}$ = 2.03 mm, GSD$_{narrow}$ = 2.02 mm). The first grain size distribution (GSD$_{broad}$) comprises a log-normal distribution from 0.25 mm to 8 mm (Figure 2). The second distribution (GSD$_{narrow}$) is only comprised of two size classes; 1.4 to 2.0 mm and 2.0 to 2.8 mm (Figure 2). As a result GSD$_{broad}$ has a substantially higher $D_{84}$ and standard deviation ($\sigma$), as would be expected from its substantially coarser and finer tails.

At the beginning of the experiment, roughness elements were placed on the bed to ensure that the flow remained subcritical during the initial deposit building stages. Once the sediment feed and water supply were turned on, bed material deposited around the initial roughness elements, burying them and creating a freely adjustable self-formed deposit with a configuration dictated by the grain size distribution of the sediment supply. The data presented here is collected after sediment has begun to exit the flume. That is, sediment has deposited along the length of the flume, sediment transport out of the flume has begun

and the sediment trap is collecting this output (see Figure 3). By which time the channel has a self adjusted roughness and the

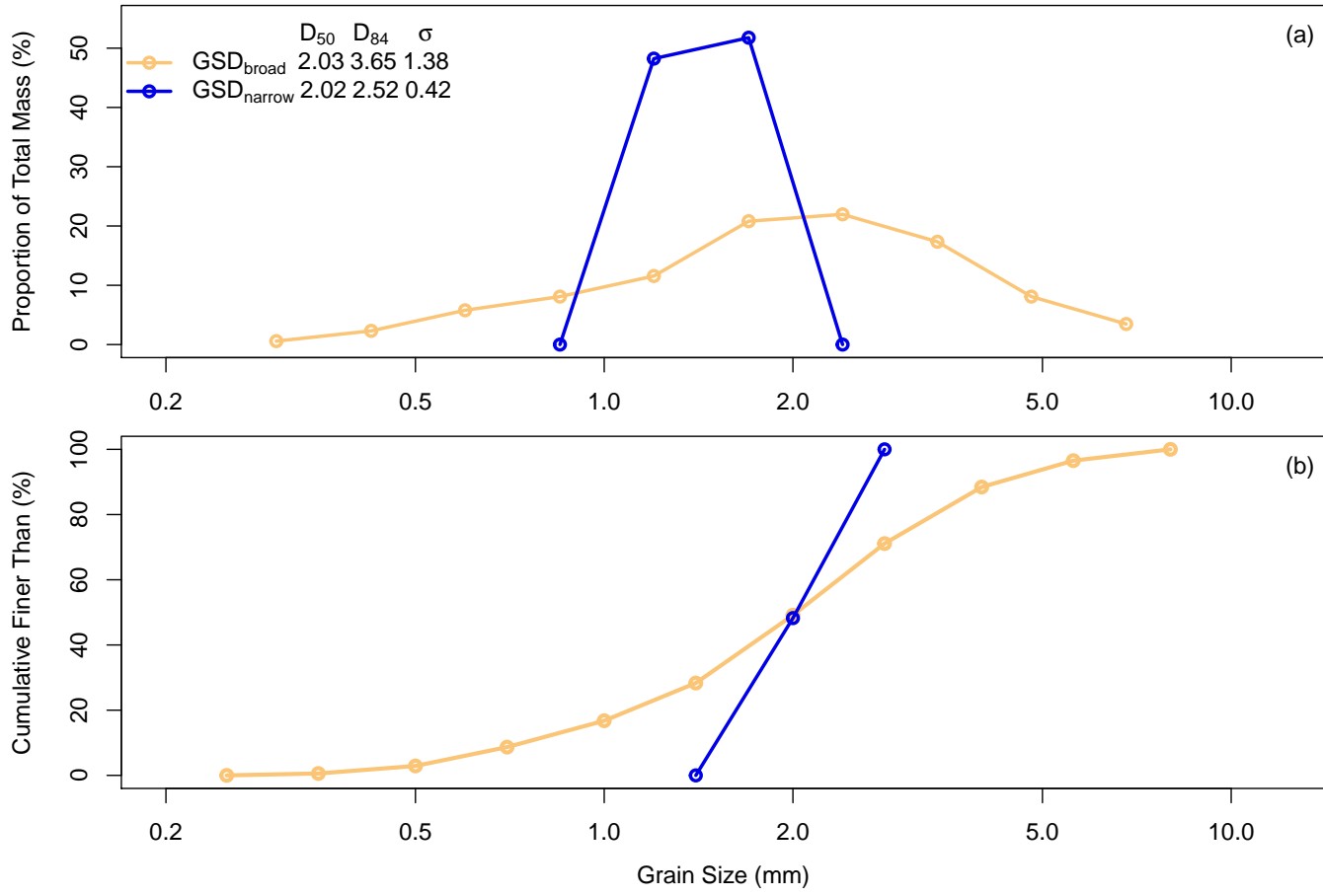

**Figure 2.** Grain size distributions for half-phi classes (a) individually and (b) cumulatively. Grain size metrics are shown in mm.

influence of the roughness elements themselves is limited. Data collection ended when the supply of sediment was exhausted, thus run time is proportional to the feed rate.

The main source of data used in this study was collected using a side-looking camera to map the evolution of the channel's long profile. A Mako Optic camera was positioned perpendicular to flume orientation, and took photographs at 60 second

5 intervals. The camera itself contains a routine to flatten these images and correct for radial lens distortion, resulting in a nearly perfect orthometric image. An image calibration routine translated pixel values to real space coordinates, from which a linear regression was fit to estimate the bulk sediment deposit gradient from the channel profile. Additionally, at 30 second intervals oblique images of the bed were captured by a GoPro oriented upstream. The images were compiled into videos and submitted alongside this paper to record the bed state evolution (see Video availability). Note that in the online videos the name $GSD_1$

10 refers to $GSD_{broad}$, and the name $GSD_2$ refers to $GSD_{narrow}$.

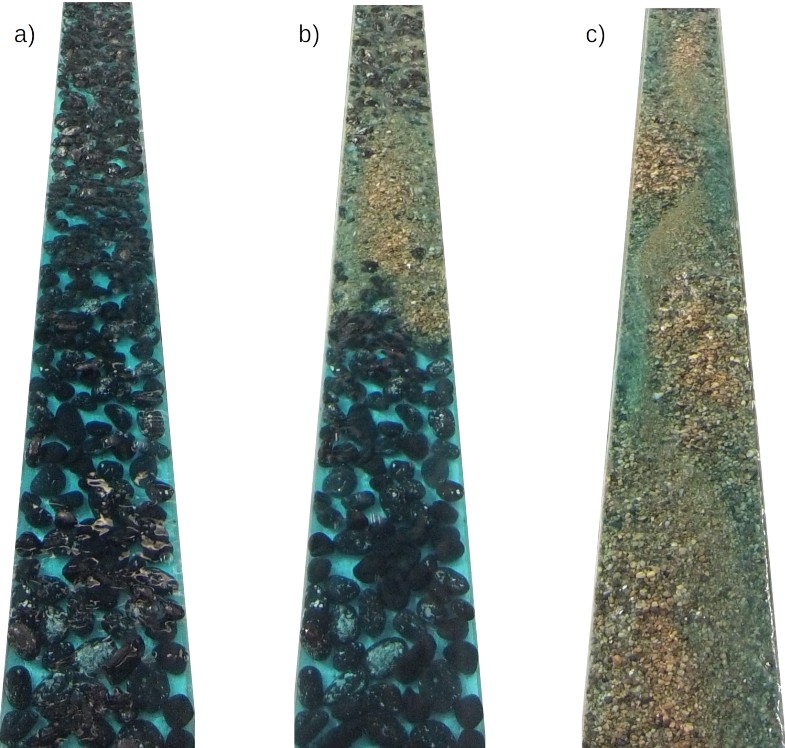

**Figure 3.** Example images of the black roughness elements used to force subcritical flow, and their submergence taken during (a) beginning of experiment with full emergence, (b) partial submergence and (c) onset of transport, almost complete burial and submergence, from experiment 200H using GSD$_{broad}$.

Sediment output data was also recorded through the use of a sediment trap emptied at 15 minute intervals. The material was captured, dried and then weighed, giving us mean transport rates for the preceding period and allowing us to calculate relative sediment storage efficiency, as the difference between output and input. That is, a relative storage efficiency of 100% represents all sediment that is input is stored during a timestep (Table 3). Additionally, the mean transport rate is the value used in Eq. (3) in order to calculate $\eta$ values reported later.

## 2.1 Slope Derivation

In order to derive a water surface slope from the profile images, a simple supervised image classification process was applied to each frame to automate the process. First, a randomForest model was used to assign one of seven sub-classes to RGB pixel values built from a smoothed training image (Liaw and Wiener, 2002). Random forests utilise decision trees, that minimise some factor, built on different, random samples of the training observations and then averaging the results of each of these decision trees to make predictions from that dataset (Breiman, 2001). Averaging the results of the myriad regressions built in the model thus improves its predictive strength, and randomForest models have been employed during supervised classification

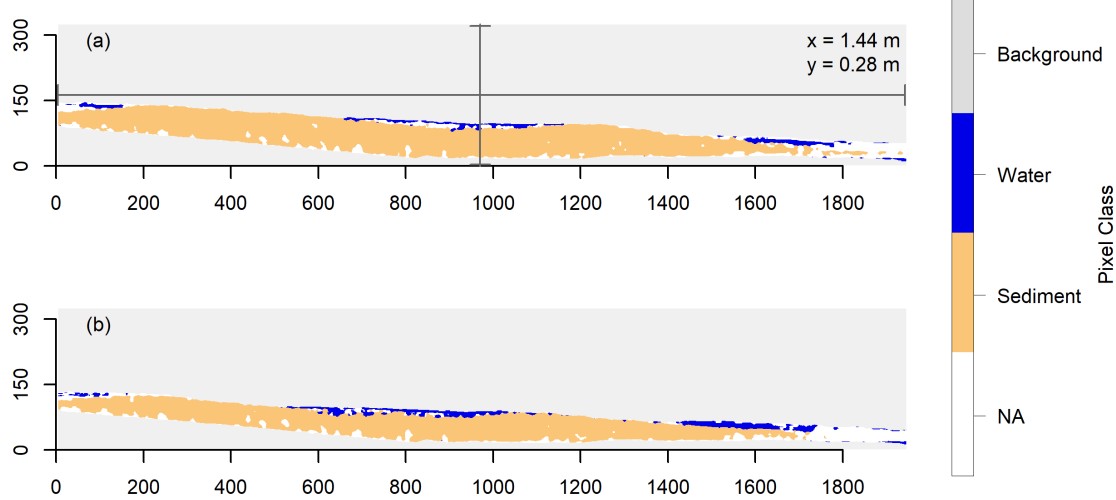

**Figure 4.** Output after image classification at the onset of output for Exp. 100L using (a) $GSD_{broad}$ (T = 326 min) and (b) $GSD_{narrow}$ (T = 149 min). Axes are pixel values of the raster, and real distances are provided for scale.

of remotely sensed images (see Belgiu and Drăguţ, 2016). The sub-classes are: sediment, clear water, water with sediment behind it (pool), water surface, background, background with shadow and the roughness elements. These sub-classes were then grouped into four umbrella classes as sediment, water, background and roughness elements, with the latter treated as NA values, and then smoothed using a 7 x 7 mode pixel filter to reduce noise (Figure 4). The training image was chosen such

5 that each of these sub-classes were present, and then a model built between the red, green and blue pixel values of each class that was applied to every other image in the dataset. The slope values are the water surface slopes defined as the boundary between background class pixels and the highest of either water or sediment class pixels, until the downstream-most extent of sediment. As sediment may infill between the roughness elements but not contiguously deposit up to that point, a manual mask was applied to the height of the roughness elements to prevent the erroneous reporting of slope values. Example slope profiles

10 show the typical calculation of the regression, at the beginning, middle and ends of runs (Figure 5).

## 3 Results

There is a substantial difference between the distribution of slopes for $GSD_{broad}$ and $GSD_{narrow}$ under 100L conditions; $GSD_{broad}$ organised to a mean 42.7% higher than $GSD_{narrow}$ (Figure 6). Similarly, for twice the relative sediment concentration (i.e., 100H), a clear separation exists between the distributions of slopes formed by $GSD_{broad}$ and $GSD_{narrow}$, albeit with

15 a lower difference between the two; the mean slope of $GSD_{broad}$ is 22.1% higher. In contrast, at higher discharges but the same relative sediment concentration (200H) the slopes for both sediment feed rates are distributed about a lower mean ($GSD_{broad}$: 0.0492 m m$^{-1}$, $GSD_{narrow}$: 0.0452 m m$^{-1}$), and substantial overlap occurs between the lower bound of $GSD_{broad}$ and the

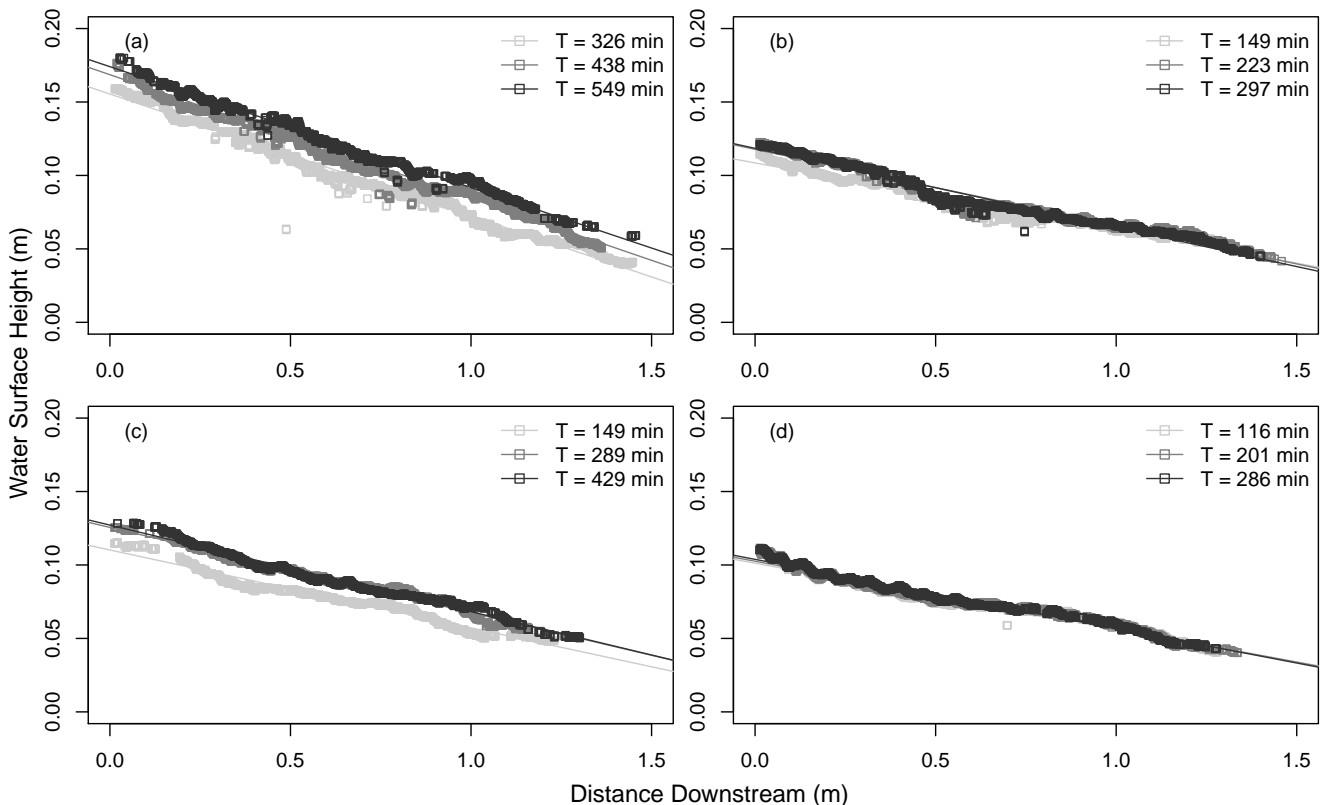

**Figure 5.** Example water surface profiles, and regressions used to derive slope, for experiments (a) $GSD_{broad}$ 100L, (b) $GSD_{broad}$ 200H, (c) $GSD_{narrow}$ 100L and (d) $GSD_{narrow}$ 200H. Times given in legend correlate to the onset of output, approximately halfway through the experiment, and the end of the experiment.

upper bound of $GSD_{narrow}$. The mean slope value decreases for both grain mixtures at lower feed rates and higher discharge (200L), although it decreases more sharply for $GSD_{broad}$ (0.0497 m m$^{-1}$) than $GSD_{narrow}$ (0.0428 m m$^{-1}$), and occupies a similar distribution as those for 200H. Mean slopes are given in Table 1, and differences resulting from changes between run conditions for the same grain size distribution are shown in Table 2.

5    Sediment output rates show a decrease in the proportion of sediment storage in response to increases in discharge (Table 3). For both grain size distributions more material is stored at lower discharges, resulting in steeper sloped deposits. Doubling the feed rate results in both systems retaining a higher proportion of sediment within the system at 100 ml s$^{-1}$, although this effect is more prominent in $GSD_{broad}$; 59.3% to 91.5%, and 27.1% to 33.6%, for $GSD_{broad}$ and $GSD_{narrow}$ respectively. However when discharge is increased, regardless of feed rate, the two systems behave more similarly both with respect to feed rate and

10   with each other. Here, 16.3% and 18.3% of material is stored for $GSD_{broad}$ under low and high feeds respectively, and only 9.4% and 8.6% for $GSD_{narrow}$. At higher discharges and higher sediment supply rates, the onset of transport occurs earlier

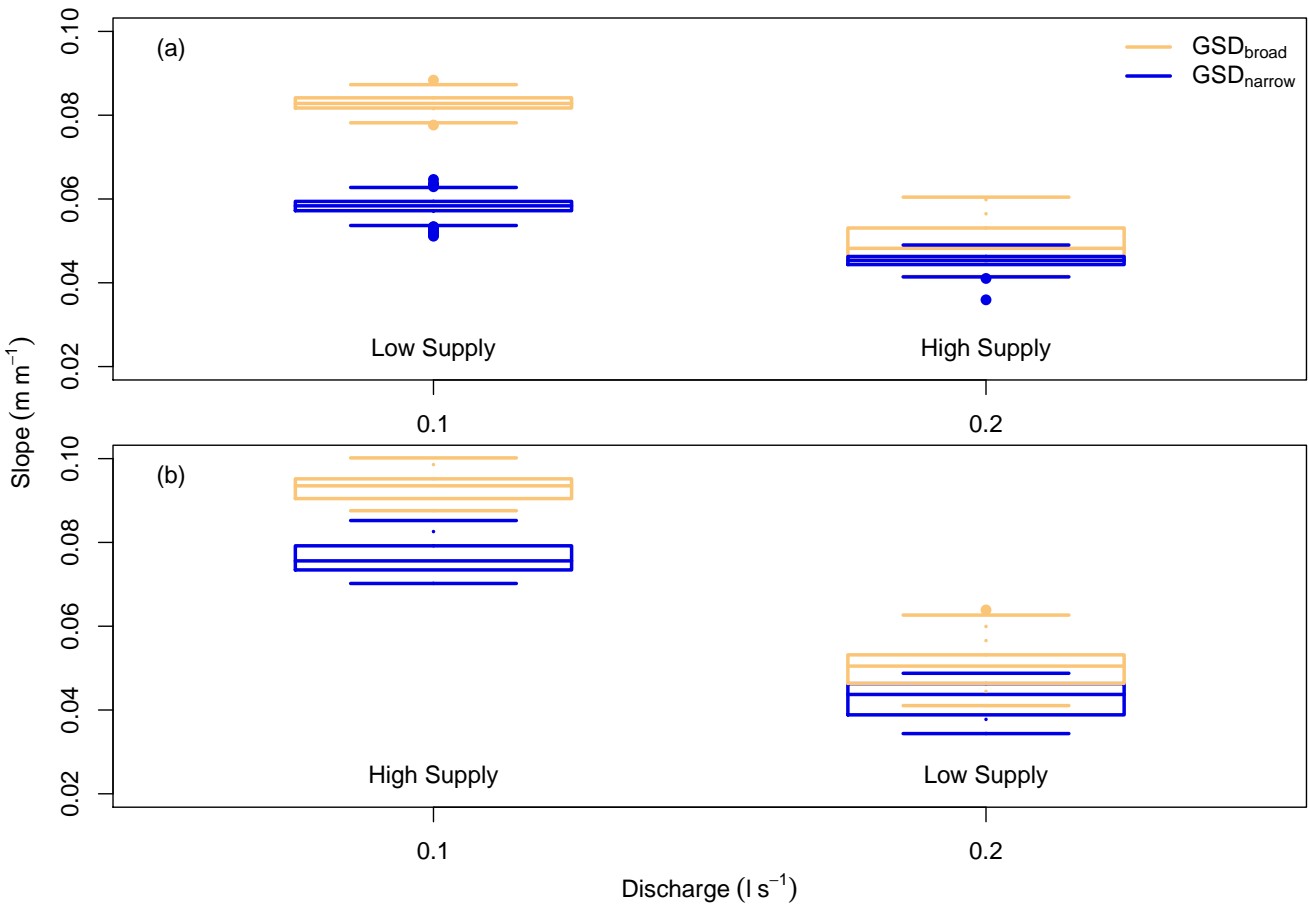

**Figure 6.** Distribution of slope values from the onset of transport onwards. Plots separate experiments by relative sediment concentrations of (a) 1 and (b) not 1 (i.e., 0.5 or 2). Text indicates sediment feed rate (low supply = 1 g s$^{-1}$, high supply = 2 g s$^{-1}$).

regardless of grain size distribution (Table 4). In addition, output starts later using GSD$_{broad}$ for all experiments barring 200H, where transport begins at almost the same time.

The distributions of the output material show a variable agreement between the input load and the output material (Figure 7 and Table 5). For GSD$_{broad}$ the $D_{50}$ of the output is coarser at all discharges, whereas the $D_{84}$ is finer at 0.1 l s$^{-1}$ and coarser at 0.2 l s$^{-1}$. The output mixture also only equals the feed $\sigma$ at high feed rate and discharge. In contrast, GSD$_{narrow}$ has a slightly higher output $D_{50}$ at low discharge and finer at high discharge, with an almost constant $D_{84}$. In addition, the $\sigma$ is always higher than the feed rate. This, caused by the addition of a fine tail, is an artefact of the rotary feeder used in these experiments; the action of the rotating feeder pipe crushed a small amount of sediment as it was input into the flume. Overall, however, there is strong agreement between the feed and output grain mixtures.

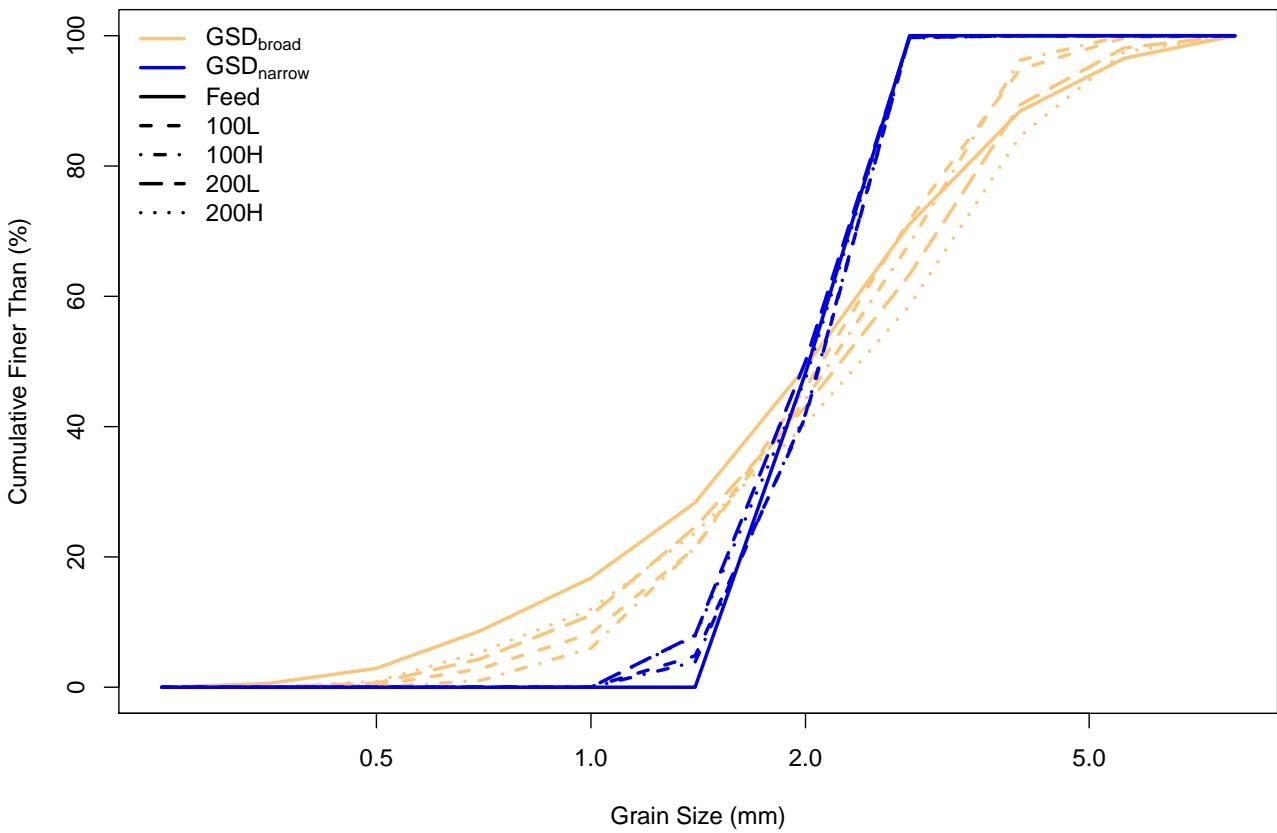

**Figure 7.** Size distributions of material output across the length of each experiment. Original grain size distributions are also provided.

Three key observations can be made regarding the distribution of transport efficiencies (Figure 8). First, the distribution of calculated transport efficiencies of $GSD_{broad}$ are consistently lower than those of $GSD_{narrow}$ (Table 6), following from the differences in slope reported above. For example, the mean transport efficiency of $GSD_{narrow}$ is 154% greater than $GSD_{broad}$ for experiment 100L. Second, increasing water discharge, for a given feed rate, increases the efficiency of both grain size
5   distributions except for $GSD_{narrow}$ under low feed, where a decrease is observed (Table 7); efficiency for $GSD_{broad}$ is 17.2% higher under 100L than 100H. Third, increasing feed rate, for a given discharge, increases the efficiency of both grain size distributions except for $GSD_{broad}$ with low discharge, where a decrease is observed (Table 7).

The mean slope values were also used to calculate the reference shear stress ($\tau_r$) necessary to entrain the $D_{84}$, using the approach of Wilcock and Crowe (2003):

$$\frac{\tau_{ri}}{\tau_{rs50}} = \left(\frac{D_i}{D_{50}}\right)^b \tag{4}$$

where $\tau_{rs50}$ is the reference stress for the median surface grain size ($D_{50}$), and b is an exponent of value 0.67 when $i$ is larger than the mean surface grain size. Equation (4) produces a shear stress 48.1% greater for entrainment of the $D_{84}$ than the entrainment of the median in $GSD_{broad}$, and 15.5% greater for $GSD_{narrow}$. This value is static for each mixture, solely based on the grains comprising the mixture and not on deposit characteristics, slope or bed state. We also calculated the mixture mobility transition point ($\tau_m^*$) from Recking (2013), which is adapted from Recking (2010), using:

$$\tau_m^* = (5S + 0.06)\left(\frac{D_{84}}{D_{50}}\right)^{4.4\sqrt{S}-1.5} \tag{5}$$

where $S$ is energy slope and the transition point represents the Shields stress where partial mobility transitions to full mobility (Table 8). $GSD_{broad}$ has substantially higher values for both low and high feeds (0.414 and 0.479 respectively) than $GSD_{narrow}$ at low discharge (0.318 and 0.417), but they decrease at higher discharges (0.228 and 0.225) and approximate those of $GSD_{narrow}$ (0.240 and 0.264), albeit slightly lower in value. That is, both mixtures exhibit similar transitions between partial and full mobility under the higher discharges, but $GSD_{narrow}$ remains substantially lower at lower discharges.

In addition, the experiments also demonstrate differences in the morphologies, particularly centred around the form and behaviour of bars in the flume. The full suite of evidence is available in the supplemental videos submitted alongside this paper, but key frames are also included here. The bars formed using $GSD_{broad}$ seem to form from the coarser end of material and exhibit greater curvature, whilst those of $GSD_{narrow}$ form bars that deflect flow to a lesser extent, that are texturally indistinguishable from the bulk mix. At lower discharges, both grain size distributions exhibit higher numbers of bars with lower wavelengths, with $GSD_{broad}$ typically organising to shorter wavelengths than $GSD_{narrow}$ (Figure 9(a) and (c)). At higher discharges, the number of bars decreases for both mixtures and their wavelengths increase to compensate, with $GSD_{broad}$ continuing to exhibit a shorter wavelength (Figure 9(b) and(d)). We also observed the occurrence of erosional events we will refer to as "thalweg sweeps", presented as a series of frames in Figure 10 and also observable in the supplemental videos. During these events the thalweg laterally erodes through the adjacent bar and then either remains on the new side, or migrates back to its original position. These bar sweeps do occur in both grain mixtures, however they are relatively limited in their frequency and magnitude in $GSD_{broad}$ and are a more defining feature of the morphodynamics of $GSD_{narrow}$.

## 4 Discussion

The results of these experiments clearly demonstrate that the range of grain sizes present in the bed material and load exerts first-order control over self-formed deposition, and it is therefore inappropriate to simply use the median surface grain size in order to characterise the system under all conditions. The extent of this influence, however, varies with the boundary conditions under which the experiments are conducted. At lower discharges, differences between the two grain size distributions can be

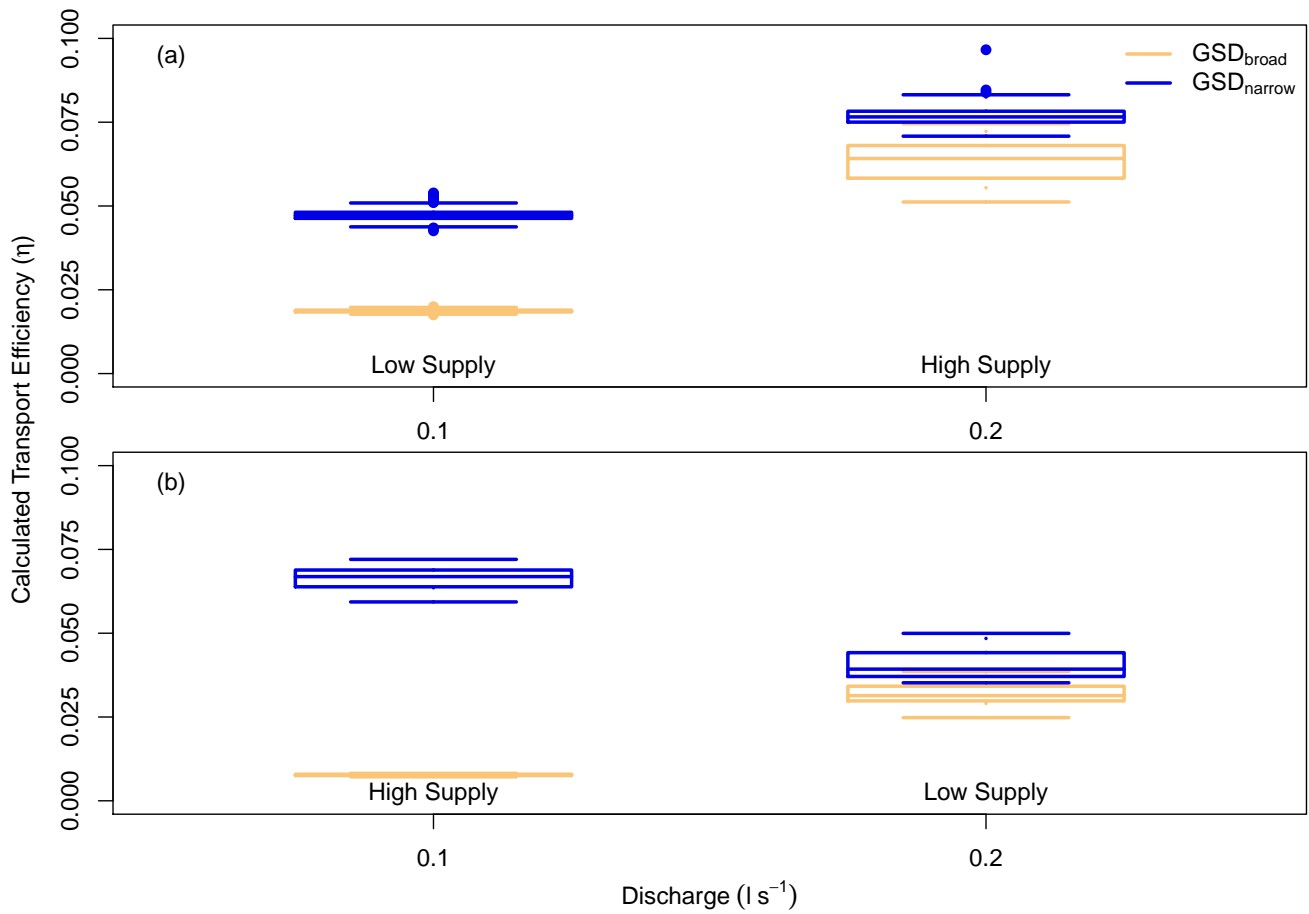

**Figure 8.** Distribution of transport efficiencies, calculated using Eq. (3). Plots separate experiments by relative sediment concentrations of (a) 1 and (b) not 1 (i.e., 0.5 or 2). Text indicates sediment feed rate (low supply = 1 g s$^{-1}$, high supply = 2 g s$^{-1}$).

attributed to the relative difficulty of the channel to mobilise the larger grains, thus it is the volume of supplied material that influences the efficiency of transport. At higher discharges, the difference in behaviour between the two mixtures decreases as the mobility differences also decrease.

### 4.1 Differential Mobility

5 The difference in mobility varying alongside discharge is shown by our primary response variable, slope. Slope acts as an indicator of system's ability to transport the material supplied to it, as mediated by the energy supplied to it (Mackin, 1948; Lane, 1955; Church, 2006; Eaton and Church, 2011). If we were to predict behaviour of the systems from the Lane and Church relations (Eq. (1) and Eq. (2)), we would assume that both grain size distributions would behave in the same manner.

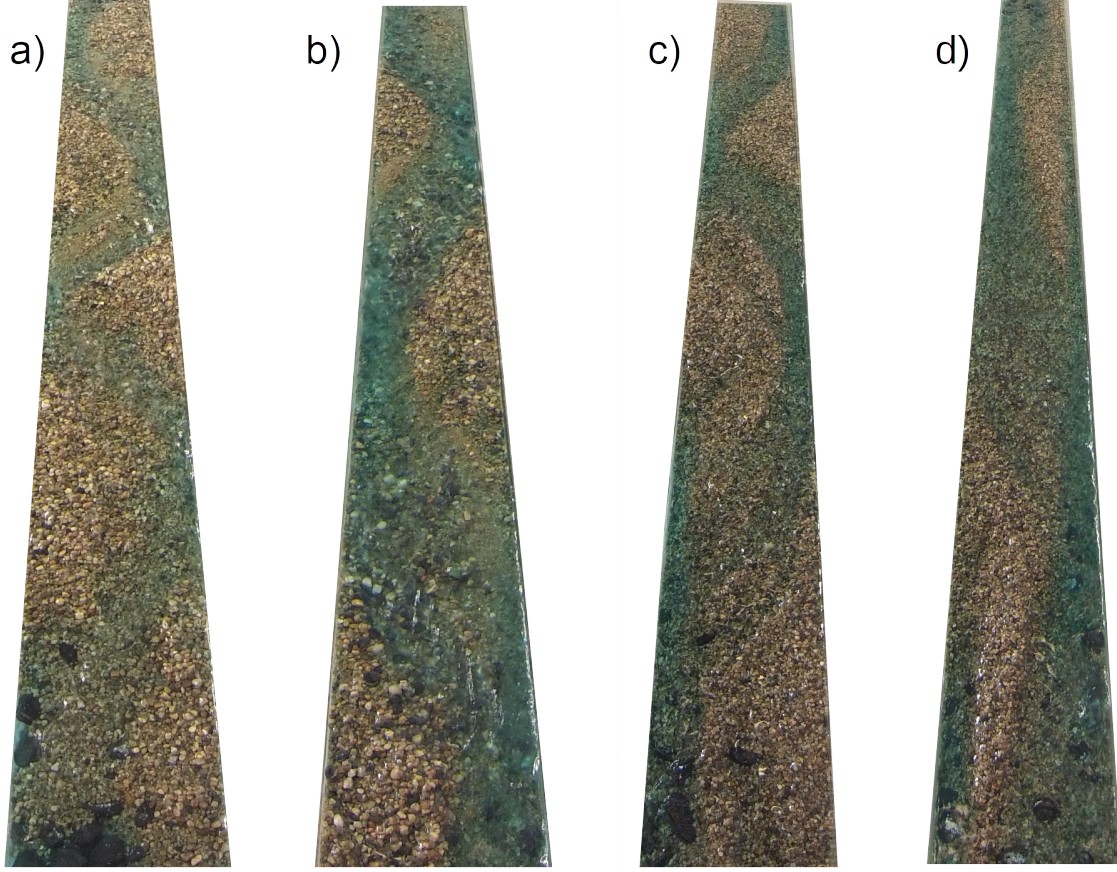

**Figure 9.** Example images of bed state taken from GSD$_{broad}$ at low (a) and high (b) discharge, and GSD$_{narrow}$ at low (c) and high (d) discharge.

Additionally, those experiments of the same relative sediment concentration (100L and 200H) would have the same values of slope. Further, where sediment concentration was increased or decreased, we would expect commensurate increases or decreases in slope respectively. However, one set of systems (i.e., those of the broadly graded GSD$_{broad}$) consistently organise to higher slopes and lower transport efficiencies than those for the more narrowly graded (i.e., GSD$_{narrow}$) systems for each experimental condition. As all systems were continuously accumulating, static sediment surface armouring (e.g., Sutherland, 1987; Parker and Sutherland, 1990; Gomez, 1994) could not occur due to the suppression of selective transport and subsequent equivalence between the bed surface and sediment feed grain size distributions. Instead, here the bed surface resembled the bed states that Iseya and Ikeda (1987) and Lisle et al. (1991) observed, in which the mixture is laterally organised. Bennett and Bridge (1995) also observed lack of bed texture adjustments under aggrading settings, when the accumulation is induced either through flume slope or feed rate changes. Therefore, we believe that the observed differences in slope cannot be attributed to differing degrees of surface armouring across the bed surface.

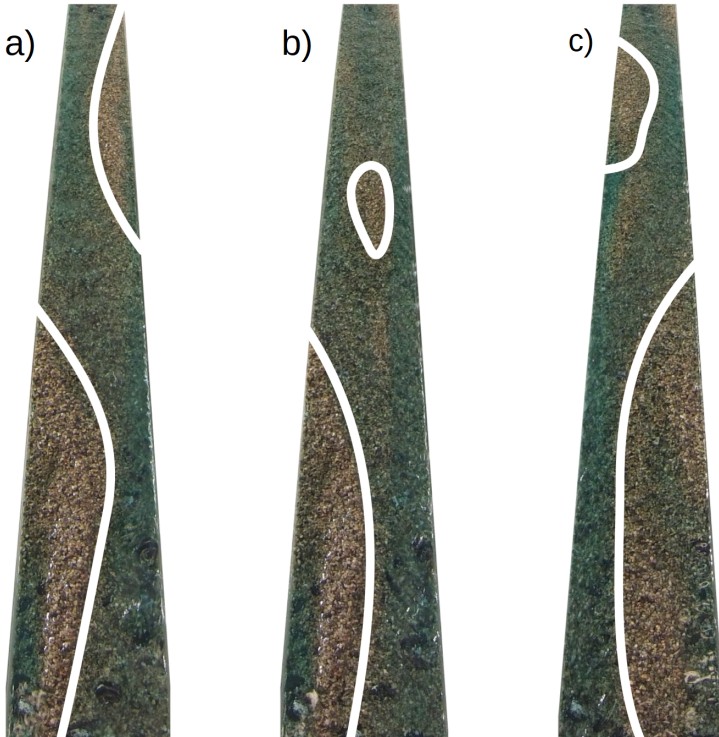

**Figure 10.** Three frames taken from the beginning, middle and end of a thalweg sweep event from the left side of the flume to the right and resulting in a switching of the thalweg position, in experiment 200H using $\text{GSD}_{narrow}$. These frames correspond to approximately 28 s to 30 s of the accompanying video.

A potentially critical explanation for this disparity is that equal mobility does not apply to all of the bed sediment, all of the time. For example, Andrews (1983) found equal mobility applied only to sediment finer than about the bed surface $D_{84}$ in his field study, and nearly all of the data on bed mobility published by Haschenburger and Wilcock (2003) showed similar relative stability of the largest grains at even the highest shear stresses. Mixtures do approach equalised mobility under higher

5  entraining stresses, although larger grains are still entrained less frequently and travel less distance (Church and Hassan, 1992). We believe that this suggests that the size of the largest sediment in the bed may determine the deposition threshold for a mixture, at least for those situations in which competence controls sediment deposition, not sediment transport capacity. The implication of which is that the gradient of self-formed, alluvial deposits is likely to be related to the mobility of the largest sediment in transport not the median grain size, as evidenced in bimodal fan experiments (Reitz and Jerolmack, 2012; Delorme

10  et al., 2017).

According to the conventions established by Lane (1955) and Church (2006), both of our grain size distributions had the same sediment calibre, so why did they not equilibrate at the same slope, and achieve the same transport efficiency? The average size of the sediment feed calibre is almost identical for both $\text{GSD}_{broad}$ and $\text{GSD}_{narrow}$. While we can explain the

failure of Eq. (1) as stemming from its original intention to be used as a qualitative guide for thinking about channel grade, Eq. (2) is based on the existing semi-empirical representations of bed sediment entrainment, so the discrepancy between Eq. (2) and the results in Figure 8 point to a more fundamental problem. Under these conditions, both mixtures do not exhibit the same mixture mobility because of the distributions' relationship with the threshold of motion; both mixtures are not equally mobile irrespective of discharge. Simply put, these results clearly indicate that $D_{50}$ is a poor choice for the characteristic grain size, at least when considering the processes forming (i.e., aggrading) alluvial deposits at lower discharges (i.e., the majority of the time), rather than those eroding them or under equlibrium conditions. The use of these equations confirms that the $D_{50}$ is only appropriate under equal mobility conditions. Our preliminary analysis suggests that some representation of the coarse tail is probably more appropriate (such as the $D_{84}$, which is commonly used in flow resistance equations (e.g., Lenzi et al., 2006; Ferguson, 2007; Recking et al., 2008)), under the partial mobility conditions shown during some of these experiments.

At a basic level, the observed difference in slopes is associated with the differential ability of two experimental systems to transport sediment, which in turn is related to the relative thresholds of motion of the largest grains. Curran and Wilcock (2005) observed a gravel-sand mixture organising to lower slopes in response to increased proportions of sand, implying that the higher sand presence decreased the critical shear stress necessary to transport coarser material, a common feature in bed organisation studies (e.g., Iseya and Ikeda, 1987). When the bed is organised as such, the variance of force exerted upon the grain and thus the likelihood of entrainment increases (Schmeeckle et al., 2007), hence the lower deposit slope and higher mobility. Here we infer that the inverse is in operation, with the presence of coarser grains decreasing the overall transport rate by increasing the entraining stresses of the mixture in a similar manner to their behaviour in degrading settings (Church et al., 1998; MacKenzie and Eaton, 2017), antithetical to the influence of sand. That is, their presence acts in a manner similar to those in kinematic waves and traction clogs in slowing the overall bed load motion (Leopold and Wolman, 1957; Moss, 1963; Langbein and Leopold, 1966; Ashmore, 1991).

Based solely on the differences in reference stress values (Eq. (4)) we would expect a fixed difference in slopes between the two grain mixtures. However, our observations of the differential system state reponses (i.e., slope) show that the degree of this difference changes with discharge. Our calculations of the transitional Shields stress, from Recking (2013), indicates that this value changes alongside discharge, as a product of slope. The differences in slope at low discharge and similarities at high discharge indicate the importance of the absolute mobility of the bed material, and the coarser fraction, in conjunction with its relative mobility. That is, if the material is subjected to a larger fluid force (i.e., higher discharge) it performs the same role as removing those grains which cause the immobility, and hence a convergence in behaviour.

The threshold calculation used in Eq. (5) invokes the partial and full mobility conditions under differing flow strengths by Wilcock and McArdell (1993). Although we cannot calculate a shear stress, given the lack of water depths, it is a useful indicator of the state of the system at a more generic scale, regardless of the actual values of $\tau_m^*$. This separation between transport regimes at low and high discharges is also similar to the observation of the three phase transport models of Ashworth and Ferguson (1989) and Warburton (1992), where full mobility is achieved above a threshold discharge following the cessation of a given influence. However, the increased transport rates are not generated through the destruction of previously organised structures (e.g., Laronne and Carson, 1976; Cudden and Hoey, 2003; Recking et al., 2009). Instead the difference is sourced

from an increase in the maximum grain size entrainable by the flow, and the likelihood of that grain's entrainment. That we see a broadly graded mixture ($GSD_{broad}$) acting in a manner similar to one that is narrowly graded ($GSD_{narrow}$) at higher discharges suggests that Eq. (2) is applicable when there is equal sediment mobility as the characteristic grain size approaches the median.

We also observed two further phenomena that may contribute to the observed output distributions. First, the finest material would often be found at the base of the flume during preparation for the next run, having filtered through the coarser matrix; a phenomenon limited to $GSD_{broad}$. The hiding of this finer material through vertical sorting explains the observed differences in the fine tail (Figure 7) as well as the constant coarser $D_{50}$ for $GSD_{broad}$. Second, there was a degree of coarse material deposition at the mouth of the feeder. However, as shown by the similarities of the output and input $D_{84}$ (Table 5), this only

affected $GSD_{broad}$ at the low discharges. Presumably, when these grains were not mobile throughout the mixture regardless of their deposition upstream. Therefore, we believe the mobility differences are systematic between the two distributions.

## 4.2 Bed Form and Dynamics

The differences in morphodynamics extend beyond reach average, 1D parameters like depositional slope and transport efficiency. Our observations of the bed dynamics have highlighted the important role that surficial organisation plays in controlling

channel morphology and influencing sediment transport rates. Surface organisation is a frequent response of channels to increased sediment supply in order to maintain some sediment coherency (Lisle and Hilton, 1992; Kasai et al., 2004; Pryor et al., 2011). The alternate bar morphodynamics we observed during some runs have been previously observed, arising from inequal stresses across the bed (Lanzoni, 2000). Under a similar grain size distribution to $GSD_{broad}$, Lisle et al. (1991) observed the formation of stationary (non-migrating) lateral bars with the bed surface separated into congested and smooth zones,

influencing transport paths.

Bed forms can influence sediment transport efficiency through the dissipation of energy and increased channel stability (Cherkauer, 1973; Hey, 1988; Prancevic and Lamb, 2015), and the bar characteristics are strongly linked to the maximum size of sediment in the bed material here. In $GSD_{broad}$, bars were more persistent in time and space than in $GSD_{narrow}$ due to the importance of large grains as stabilising features for bars. In the case of $GSD_{broad}$ the largest grains clearly deposited

first, creating a locus of deposition around which the bar head formed, allowing additional sediment to accumulate in its wake (Leopold and Wolman, 1957; Ashmore, 1991; Ferguson, 1993). The bars formed during $GSD_{narrow}$ were comprised of virtually the same size sediment, which can be entrained over a narrow range of shear stresses. As a result, the whole bar may be entrained at a similar shear stress, making these features more transient, and reducing their overall effect on bed stability as their relative impermanence means that the flow can freely move through them (Figures 9 and 10). It is important to note

that the stabilisation of the large grains in our experiment $GSD_{broad}$ was not the result of jamming, as described by Church (2006). The size of our flume was such that the ratio of the flume width to $D_{84}$ was greater than 6, the jamming ratio proposed by Zimmermann et al. (2010), and is thus solely the result of deposition.

The formation of lateral bars allows the transport of bed load through the contraction of the channel width increasing unit stream power as flow is concentrated (Lisle, 1987). This organisation of the bed surface into zones of transport and deposition

thus maximises the efficiency of the channel (Iseya and Ikeda, 1987; Ferguson et al., 1989), and enables the previously limited transmittance of sediment and the growth of the depositional lobe. The wavelengths of the barforms we observed in higher sediment output experiments are longer than those of high sediment storage. Experiments by Pyrce and Ashmore (2003a) and Pyrce and Ashmore (2005), and a meta-analysis by Pyrce and Ashmore (2003b), demonstrated that the wavelength of

bar spacing is a function of the transport lengths of bed load particles at channel forming flows, therefore the material is more mobile in higher wavelength reaches. Transport length is the distance between entrainment ($\tau_{ce}$) and distrainment ($\tau_{cd}$), therefore it is dependent on when the grain is deposited. Ancey et al. (2002) observed a type of hysteretic difference between these two thresholds, where the specific flow rate (and thus stress) necessary to induce deposition is lower than the entraining flow. Given the differences in entraining threshold between the mixtures, assuming that it is controlled by the coarse tail,

it follows that the distraining threshold for $\text{GSD}_{broad}$ will be higher than for $\text{GSD}_{narrow}$, such that a smaller decrease is needed to trigger deposition. Therefore the systems can be characterised by the difference in behaviour of the coarse grains comprising the bar head loci (Pyrce and Ashmore, 2005). For the more equally mobile $\text{GSD}_{narrow}$ this is manifested in a decreased likelihood of deposition of these grains, triggering longer path lengths and greater bar wavelengths in the system. In other words, the likelihood of entrainment ($P[\tau > \tau_{ce}]$) is greater in $\text{GSD}_{narrow}$, setting a lower overall deposit slope, whereas

the likelihood of distrainment ($P[\tau < \tau_{cd}]$) is higher in $\text{GSD}_{broad}$, decreasing transport length. This difference in reduced as discharge increases because the likelihood of entrainment increases, and distrainment decreases, for both mixtures but more strongly for $\text{GSD}_{broad}$; hence the similarity between $\tau_m^*$ values at higher discharges.

Were we to pinpoint the actual characteristic grain size of the material, we might expect the slopes to actually organise to the same values. For example, if we were to pair these distributions instead by a coarser grain (e.g., the $D_{84}$) we might

have observed more similar self-organised slopes. However, this view still assumes the same inherent grain class mobility across discharges, that is the theoretical basis behind Church (2006), merely shifted in favour of the larger grains contributing more relative stability. That is, each mixture still has different distributions of transport likelihoods; does the characteristic grain size actually represent enough of the bed processes that the overall system behaves in the same manner? If the lateral bars were composed of coarser material, with the same narrow gradation as $\text{GSD}_{narrow}$, the depositional slope and wider

morphodynamics may be similar (i.e., general organisation), but the finer scale processes (i.e., sediment transport and meander wavelength) would not be. However, it might be the case that similar characteristic grain sizes are just an artefact of the experimental design, rendered irrelevant when boundary conditions are expanded to different ranges. Additionally, mobility differences can be more strongly controlled through channel widening and planform adjustments than allowed within this flume, thus the importance of grain class thresholds are reduced as there are more options for resistance to be generated (Eaton

and Church, 2004). Thus, the discussion of any one characteristic size is only useful within a given comparison, and does not necessarily indicate a behaviour fundamental to self-formed channels but is merely the smallest partially mobile grain class (Wilcock, 1993).

We can therefore consider the systems generated by $\text{GSD}_{narrow}$ as less stable on three accounts. Firstly, they developed at lower slopes and, as a result, were able to prograde more quickly due to reduced deposit volume. Secondly, the grains were

more equally mobile due to a lower maximum threshold stress (i.e., smaller coarse fraction). Thirdly, the degree of surficial

organisation was lower and bedforms were less persistent. The combination of these factors results in a system that does not need to concentrate flow in order to exceed threshold stress, despite the flow's lower slope and stream power, indicating its ability to transport sediment more efficiently than in experiments using $\text{GSD}_{broad}$.

## 5  Conclusions

The eight experiments presented here demonstrate a difference in the self-adjusted slope and morphodynamics of aggrading systems derived from the difference of their grain size distributions and mediated by the relative sediment concentration. According to the prevailing theory that the median grain size is predictive of channel behaviour, the systems described within this paper should have exhibited similar slopes and patterns of morphodynamics under the same boundary conditions. Instead, the deposit formed from the more widely graded distribution ($\text{GSD}_{broad}$) developed to a higher slope, with lower transport

efficiency, and demonstrated a greater degree of surface organisation. We argue that this is the result of the large grains present in this mixture that exceed the competence of the flow, and require channel narrowing in order to mobilise. Where these grains are absent (i.e., the narrowly graded $\text{GSD}_{narrow}$) the channel fails to organise to its most stable state (i.e., lateral bars with narrow thalweg) as regularly because of the more equally mobile sediment and bars. This difference decreases as discharge is increased (i.e., entraining stresses). Thus channel stability is linked not to the mobility of the median grain size, but to the

mobility of the largest grains (e.g. $D_{84}$). We therefore conclude that the difference in behaviour between these systems is driven by a competence limitation of the larger grains. These findings indicate that models that include sediment transport and conceptualise stability, such as regime models, need to consider the characteristic grain size as a coarser fraction than the median in order to more realistically replicate behaviour in aggrading systems.

*Code and data availability.*  Both the code and data used to create Figures 2, 4, 5, 6, 7 and 8 are available online (http://doi.org/10.5281/zenodo.2672918).

*Video supplement.*  Videos of the bed morphodynamics are available online; $\text{GSD}_1$ corresponds to $\text{GSD}_{broad}$: Q100L (http://doi.org/10.5446/41771), Q200H (http://doi.org/10.5446/41772), Q100H (http://doi.org/10.5446/41773), Q200L (http://doi.org/10.5446/41774). $\text{GSD}_2$ corresponds to $\text{GSD}_{narrow}$: Q100L (http://doi.org/10.5446/41775), Q200H (http://doi.org/10.5446/41776), Q100H (http://doi.org/10.5446/41778 Q200L (http://doi.org/10.5446/41777).

*Author contributions.*  W.H. Booker collected and analysed the code and data used in the paper, drafted the manuscript, and created the

figures and tables; B.C. Eaton organised, reviewed and edited the manuscript.

*Competing interests.*  The authors declare that they have no conflict of interest.

*Acknowledgements.* The authors would like to thank the reviewers for their comments and time, which we think have greatly improved the content and character of this paper. We would also like to thank Jens Turowski and the editorial board for their help. We are thankful to Lucy MacKenzie, David Adams and Anya Leenman for the useful and insightful discussions had regarding this work.

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

**Table 1.** Mean slope values and standard deviations, for the eight experiments reported here.

| | $\mathrm{GSD}_{broad}$ | | | $\mathrm{GSD}_{narrow}$ | | |
|---|---|---|---|---|---|---|
| | Mean | St. Dev. | n | Mean | St. Dev. | n |
| 100L | $8.29 \times 10^{-2}$ | $1.96 \times 10^{-3}$ | 224 | $5.81 \times 10^{-2}$ | $2.21 \times 10^{-3}$ | 281 |
| 100H | $9.30 \times 10^{-2}$ | $2.99 \times 10^{-3}$ | 101 | $7.68 \times 10^{-2}$ | $3.93 \times 10^{-3}$ | 175 |
| 200L | $4.98 \times 10^{-2}$ | $4.28 \times 10^{-3}$ | 437 | $4.28 \times 10^{-2}$ | $3.89 \times 10^{-3}$ | 449 |
| 200H | $4.92 \times 10^{-2}$ | $4.36 \times 10^{-3}$ | 256 | $4.52 \times 10^{-2}$ | $1.57 \times 10^{-3}$ | 243 |

**Table 2.** Changes in mean slope for the column name experiment, given relative to the row name experiment, for $\mathrm{GSD}_{broad}$ and $\mathrm{GSD}_{narrow}$ respectively. Values are given in percent.

| | 100L | 100H | 200L | 200H |
|---|---|---|---|---|
| 100L | - | 12.2/32.1 | -40.0/-26.4 | -40.6/-22.2 |
| 100H | -10.9/-24.3 | - | -47.1/-41.1 | -46.5/-44.3 |
| 200L | 66.7/35.9 | 86.9/79.6 | - | -1.03/5.77 |
| 200H | 68.4/28.5 | 88.9/69.8 | 1.05/-5.46 | - |

**Table 3.** Output of sediment during each experiment, from the time sediment output occurred. The proportion of sediment output is relative to the volume input over the same timespan.

| | Mean Output Rate (g s$^{-1}$) | | Proportion Stored of Input (%) | |
|---|---|---|---|---|
| | $\mathrm{GSD}_{broad}$ | $\mathrm{GSD}_{narrow}$ | $\mathrm{GSD}_{broad}$ | $\mathrm{GSD}_{narrow}$ |
| 100L | 0.41 | 0.73 | 59.3 | 27.1 |
| 100H | 0.19 | 1.34 | 91.5 | 33.6 |
| 200L | 0.84 | 0.91 | 16.3 | 9.4 |
| 200H | 1.64 | 1.84 | 18.3 | 8.6 |

**Table 4.** Timing of the onset of transport, given in minutes from the start of the experiment.

|       | $\text{GSD}_{broad}$ | $\text{GSD}_{narrow}$ |
|-------|------|------|
| 100L  | 326  | 149  |
| 100H  | 149  | 116  |
| 200L  | 103  | 87   |
| 200H  | 42   | 44   |

**Table 5.** Grain size statistics of output material, averaged over the total output mass and given in mm.

|       | $\text{GSD}_{broad}$ | | | $\text{GSD}_{narrow}$ | | |
|-------|----------|----------|----------|----------|----------|----------|
|       | $D_{50}$ | $D_{84}$ | $\sigma$ | $D_{50}$ | $D_{84}$ | $\sigma$ |
| Bulk  | 2.03 | 3.65 | 1.38 | 2.02 | 2.52 | 0.42 |
| 100L  | 2.11 | 3.38 | 1.03 | 2.10 | 2.55 | 0.45 |
| 100H  | 2.17 | 3.42 | 1.00 | 2.10 | 2.55 | 0.44 |
| 200L  | 2.24 | 3.72 | 1.24 | 2.00 | 2.52 | 0.48 |
| 200H  | 2.39 | 3.94 | 1.39 | 2.03 | 2.53 | 0.48 |

**Table 6.** Mean transport efficiencies and their standard deviation with the number of observations ($n$) of the experiments.

|       | $\text{GSD}_{broad}$ | | | $\text{GSD}_{narrow}$ | | |
|-------|------|----------|-----|------|----------|-----|
|       | Mean | St. Dev. | n   | Mean | St. Dev. | n   |
| 100L  | $1.87 \times 10^{-2}$ | $4.42 \times 10^{-4}$ | 224 | $4.75 \times 10^{-2}$ | $2.21 \times 10^{-3}$ | 281 |
| 100H  | $3.21 \times 10^{-2}$ | $2.81 \times 10^{-3}$ | 101 | $4.05 \times 10^{-2}$ | $3.82 \times 10^{-3}$ | 175 |
| 200L  | $7.71 \times 10^{-3}$ | $2.48 \times 10^{-4}$ | 437 | $6.60 \times 10^{-2}$ | $3.29 \times 10^{-3}$ | 449 |
| 200H  | $6.33 \times 10^{-2}$ | $5.50 \times 10^{-3}$ | 256 | $7.68 \times 10^{-2}$ | $2.80 \times 10^{-3}$ | 243 |

**Table 7.** Changes in mean transport efficiency for the column name experiment, given relative to the row name experiment, for $\text{GSD}_{broad}$ and $\text{GSD}_{narrow}$ respectively. Values are given in percent.

|       | 100L        | 100H        | 200L        | 200H       |
|-------|-------------|-------------|-------------|------------|
| 100L  | -           | -58.7/39.1  | 71.9/-14.7  | 239/61.9   |
| 100H  | 142/-28.1   | -           | 316/-38.6   | 721/16.4   |
| 200L  | -41.8/17.2  | -76.0/63.0  | -           | 97.3/89.8  |
| 200H  | -70.5/-38.2 | -87.8/-14.1 | -49.3/-47.3 | -          |

**Table 8.** Mixture mobility transition points calculated using Eq. (5), taken from Recking (2013).

|      | $GSD_{broad}$ | $GSD_{narrow}$ |
|------|-------|--------|
| 100L | 0.414 | 0.318 |
| 100H | 0.479 | 0.417 |
| 200L | 0.228 | 0.240 |
| 200H | 0.225 | 0.264 |

**Table 9.** Flume dimensions and run conditions for Lisle et al. (1991) and the two 100L experiments included here.

|                          | Lisle et al. (1991) | $GSD_{broad}$ | $GSD_{narrow}$ |
|--------------------------|---------------------|---------------|----------------|
| Length (m)               | 7.5                 | 2             | 2              |
| Width (m)                | 0.3                 | 0.128         | 0.128          |
| Slope (m/m)              | 0.03                | 0.083         | 0.058          |
| Grain Size Range (mm)    | 0.35-8              | 0.25-8.0      | 1.4-2.8        |
| $D_{50}$ (mm)            | 1.4                 | 2.03          | 2.02           |
| Flow Rate (ml s$^{-1}$)  | 582                 | 100           | 100            |
| Feed Rate (g s$^{-1}$)   | 8.4                 | 1             | 1              |
| Run Time (min)           | 560                 | 549           | 429            |