# Peer review of "Stabilising large grains in self-forming, steep channels"

_Earth Surface Dynamics, 2019_

## Referee Comment (RC1) · Anonymous Referee #1 · 25 Jun 2019

Dear authors,

The study presented was pretty interesting and fascinating. After reading and analyzing it several times it seems to me that the role of GSD in an aggrading channel is evident and there is solid evidence to state that. I believe that the experiments are well designed and allow the reader to make comparisons between both GSD. Furthermore, these experiments allow one to compare different boundary conditions which makes them really valuable to the community.

After careful consideration, especially taking into account the results shown here, I believe that the article needs to be re-formulated and go through major revisions. The reasons are explained in detail in the following lines but in summary, it is because of the text structure and the way how the information was conveyed. The experiments are the

strong point of the article, however, it is difficult to understand all the ideas the authors are trying to express and also how the hypothesis and initial motivation are developed throughout the text. Some statements are also unclear and will be discussed below.

General comments: 1) This a personal opinion. The article's title could be changed to something more appropriate. When I received the article I thought that it was related to large chains in the sense of boulders or macro-roughness elements. Given that most steep channels do have boulders and other (actual) large grains, and those are neglected in this study, the title was misleading to me. Again, this is a personal opinion but please consider it if you think the same.

2) The article structure does not convey the information in a fluid manner. The introduction has little information about aggrading systems and it seems to me that it gives more importance to degrading systems. Although, I understand that the idea was to make clear that we know more about degrading systems more information and references to what we really know about aggrading systems is required. There are virtually no references to any study that may have discussed aggrading systems.

3) The article presents the study using Lane (1955) balance expression. Then, the assumptions of this expression are called into question and by doing so the hypothesis is formulated. The problem is that Lane 1955 did not consider a mixture of sediment and therefore does not intended to explain the responses of different GSD, even when they have the same D50. Only later in the paper, in the discussion (page 10, line 4), this is explained. So, as a reader, I had problems trying to understand why this is not explained right away. The major concern about this is how the information flows in the article.

4) The hypothesis needs to be reformulated. I understand the idea of the study is to compare responses to different GSD and boundary conditions. This was well developed in the text. However, if I just take the hypothesis, it doesn't say that. "We hypothesise that, like degrading systems, the presence of the large grains will result

in different transport regimes, as in MacKenzie and Eaton (2017), and thus different channel morphodynamics and depositional slope" It says that it is just the presence of large grains, what about boundary conditions? The article shows that is not just the presence of these large grains but discharge is a fundamental control.

5) A lot of information about bed structure, for example bars, is given by the end of the discussion. There is no data about this and only observations. This should be presented in a more formal way.

I belive that all these comments can be easily addressed by changing the text structure.

Specific Comments:

1) Abstract - line 2 - there is no need to say "shape", it is already included in the distribution. 2) Abstract - line 4 - Is it correct to talk about "fan" if we are in a 1D system? The fan part is where the system spreads and here it does not occur. 3) Introduction - line 11 - There is one problem when we use the discharge as a variable to explain a certain response. If we double (or 3X, 4X, ...) the channel width while holding the discharge we may have different geomorphological responses. Therefore, is not actually the discharge, but some other characteristic (e.g., unit discharge) what is better for comparisons. This may be discussed somewhere. 4) Intro - line 11 - There are several other references to this statement (sed supply and discharge controls) 5) Intro - Most of the Intro - Generally only one reference is given for a certain statement. More references are required. For example, when talking about armour layers (line 24) only Andrews and Parker, 1987 is cited. 6) Intro - Page 2 - Line 8 - References are needed for this statement. 7) Intro - Page 2 - Line 9 - Would it be better to start the discussion with something relatively newer than Lane (1955)? The experiments are really interesting, but starting the analysis based on this relatively old study (there is more information available related to stream power). It doesn't mean that this expression is not important, but, it does not fit what we know about sediment mixtures. 8) Intro - Page 2 - Line 21 - The text is confusing. However what? Please notice that the idea does

not flow starting with "however". There are some equations, definitions, and other text that makes this "however" confusing. 9) Intro - Page 2 - Line 23 - This is critical, Lane never said that this works for a sediment mixture, as you mentioned in the discussion. Therefore, up to this line, calling into question the assumption is not valid. please try to find another way to present the hypothesis. 10) Methods - general - This is the strong part of the article, it was really interesting. 11) Methods - general - It would be really interesting to analyze the evolution of the slope, that is, change of slope in time. I was wondering if the experiment came to a final equilibrium slope, or how do you decide to finish an experiment. Do we find the mean slope by the end of the experiment or by the middle of it. A simple plot would answer these interesting questions. 12) Results - page 6 - line 8 - This statement is only true for 0.1 ml sˆ-1. Notice that in panel a) for 0.2 ml sˆ-1, there is no "strong distinction". 13) Results - page 7 - line 3 - Notice that you need to reference Table 5 when you talk about the efficiency. 13) Results - General - It would be good to have more information about the properties of the bars that are mentioned at the end of the discussion. 14) Discussion - General - Some parts can be moved to the intro for a better motivation for the study. Also, it would help understand the hypothesis. 15) Discussion - General - Like in the intro, more references are needed. It is generally poor in important references. 16) Discussion - General - I'm not making a lot of detailed comments in the discussion because it seems to me that in the new version it will change significantly. Only the most important specific points are considered here. 17) Discussion - Page 9 - Line 30 - It seems that Church's relation can be a better way to motivate the study. 18) Discussion - Page 10 - Line 10 - It would be interesting to consider a little discussion about what may happen if we have the same D84 and different D50. 19) Discussion - Page 10 - Lines 12 to 17 - These lines are confusing. First, you mentioned that in low slopes sand plays an important role and that you can make the same inference. Then you said it is not actually sand what is the control in your experiments but the absence of large grains. Notice that your statement is correct (it is the absence of large grains), but relating it to Curran and Wilcock does not make any sense, because they attributed to sand. 20) Discussion - Page 10 -

**ESurfD**

Interactive
comment

Lines 20 to 32 - A lot of confusing statements are given here. a) One important aspect that you are considering is channel slope. The analysis made using Eq. 4 does not include channel slope, even though it is known that slope plays a role critical shear stress Lamb has published a number of studies related to this topic. b) Comparing a change in critical stress change for D84 to a change in slope is confusing. Why can we do that? The problem is that for a given discharge if we vary slope water depth changes as well, therefore changes in slope and not directly comparable to changes in shear stresses. Maybe I'm missing something but if you explain a little more about this rationale it would be clearer. 21) Discussion - Everything related to bars and beyond reach average - Most of the text is not clearly related to data or measurements. It need to be better justified. 22) Discussion - Page 12 - Line 6 - There are two more (more more mobile) 23) Conclusion - Page 12 - Line 34 - Change you in " as you increase". Also a period is missing.

Again, as I said before, the study is quite interesting. I hope you can address these comments and I hope that these observations and comment can improve the text.

---

## Referee Comment (RC2) · Anonymous Referee #2 · 27 Jun 2019

Booker and Eaton presents a set of laboratory flume experiments aimed at determining the role of the full sediment grain size distribution, rather than the more commonly-used median grain size, in capturing the behavior of aggrading alluvial channels. Through this targeted series of experiments, the authors find that the equilibrium slope of the aggrading channels that they produce is dependent on the grain size distribution of supplied sediment, the rate at which it is supplied, and the discharge available to transport the supplied sediment. Overall, I found this to be a nice set of experiments and a compelling result highlighting the role of bed structure in determining the dynamics of aggrading alluvial channels. While I think these results would be of interest to the community, below I have outlined a few points that I think should be clarified prior to publication.

[Figure]

• Reorganization of introduction - I think the introduction reads fairly well, but that further motivation could be provided by discussing the predictions of Lane's balance at the beginning of the article. One could use the idea that Lane's balance would predict the same slope for a give D50, regardless of the rest of the GSD as a null hypothesis, then reference the known importance of large grains in degrading systems and the lack of complementary work on aggrading systems in order to more directly motivate this work. I think this reorganization could help to streamline the logical progression of the manuscript.

• Methods clarification - While I generally follow the experimental set-up, I think some more detail can be provided regarding a few points.

(1) How where the discharges determined? Are they specified to span the range of partial transport to full bed mobilization? It would also be useful to provide the calculated/estimated shear (or Shields) stresses related to each of these discharges of both flows. I'm aware that this may require some assumptions in relation to the sidewall correction, but given that most of the literature on this topic is presented in terms of Shields stress, it would be useful to also provide this estimate, especially for the discussion of relative transport capacity. (2) It took me until halfway through the results to recognize that the multiple measures of slope presented were from different time steps following the onset of sediment transport out of the flume. How long were the experiments run after this point and how were the experiments determined to be over? Was an equilibrium slope/transport rate reached or were adjustments still occurring when the experiment ended? If equilibrium was reached, how was it determined? (3) For the slope-derivation, I think more information should be provided regarding the randomForests model, how it works, and the degree of user-specification it requires. How many images are input in order to determine the slope? How are the sub-classes determined? Are there uncertainties associated with these slope measurements based on the method or number of sample images input? A citation here providing the relevant background information could also help. The authors later report the mean slope

and standard deviation for each experiment, but it is unclear if this is from multiple time slices (if so, how many?), multiple locations in the flume, or related to some uncertainty in the slope estimation. Organization-wise, I don't necessarily think this needs its own section in the methods. Alternatively, I might suggest splitting the methods section into (1) Experimental set-up, (2) Measurements, and (3) Slope derivation. ' (4) I find GS1 and GS2 not to be very informative variable names. I would suggest changing them to GSnarrow and GSbroad or something more information so it is easier for the reader to keep track of throughout the paper. Even H and L are a bit confusing to keep track of, but less so.

• Organization of the results section - I found this section to be a bit muddy, with parts of the motivation, methods, and discussion being mixed in. While I am okay with some intermingling of these sections, in this cas, I found it to make this particular section a bit difficult to follow. Below I've made some suggestions to streamline this section. (1) Move Lane's balance discussion to introduction. See above. (2) Move sediment transport efficiency calculation to methods. I would suggest adding this following the slope derivation. If Lane's balance has already been presented in the introduction, it would naturally follow to calculate the sediment transport efficiency. Introduction of this calculation in the methods would allow the authors to more cleanly step through the results. Again, some information of the number of samples used to make these calculations would be helpful (table 5). (3) This is a style thing, but I would suggest avoiding things like "Panel A of Figure 3 shows.." and instead simply say "There is a significant difference between equilibrium slopes as a function of the supplied grain size distribution (Figure 3A)." I think this would help with readability. (4) Much of the information in the tables is not fully presented in the paper. I would recommend more explicitly discussing these results in the main text. Lots of the results are presented in a fairly vague way (e.g. - "...both systems retaining a higher proportion of sediment" even though the authors have quantified these effects more directly. I would suggest rephrasing to provide these values directly in the text (e.g. – "... in response to a doubling of sediment supply, both systems retained a higher proportion of sediment,

XX% for the narrow GSD and XX% for the broad GSD." This in-text quantification would also help to clarify the main differences between the experiments.

• Argument for large grains – While I find the argument that the transition between partial transport and full mobilization of the GSD drives the observed differences in slopes observed in the experiments reasonable, I am not entirely convinced that the data presented really show this. I agree given the results that D50 is a poor metric for predicting behavior in aggradation systems, but I think more could be done to support the argument of the importance of large grains.

Do the authors have any observations from the experiments to be show this? For example, was the sediment exiting the flume sieved to determine the GSD of the transported sediment compared to the supplied sediment? Can the photos/videos of the bed be used to determine if there is significant sorting that arises during the experiments that may support this idea? I imagine that the videos could be used to track the mobility (or immobility) of the largest grains (or the bed surface as a whole) in the flume to better evaluate this idea.

The portion of the discussion where shear stress calculations are made is quite confusing. It is unclear what inputs are being used and what information is being drawn from the calculation. Specifically, this sentence is quite unclear "Equation 4 produces a shear stress 44.4% greater for entrainment of the D84 than the entrainment of the median in GSD1 than in GSD2." I assume the authors are solving for tau_ri with reference to the D84 of both GSDs, but the reference stress value and the actual calculated values should be made explicit to better support this point. Additionally here, a comparison to the estimated shear (shields) stresses applied in the experiments (see previous comment) would help to bolster this point.

• Discussion of bar formation and effects – Currently, I think this point of the discussion appears as an afterthought. While I agree that this might not be the main result of the paper, the authors describe the differences in bar presence and morphology between GS1 and GS2 experiments in order to support their conclusions regarding the role of large grains. If this is a main point to bolster the argument related to the importance of large grains, mapping of these bar formations and quantifying their differences between runs should be included in the methods/results sections of the manuscript. This discussion would be better supported with photos or measurements in the text to more clearly illustrate the argument made.

Figure comments: General – Yellow is difficult to see, consider changing.

Figure 1 – Provide flume dimensions Figure 2 – on plot report D50, D84, and sigma as part of the legend (eliminates the need for Table 1) Figure 3 – Could combine with Figure 1? I'm not sure this particular image adds very much. Also revise run name fro G2Q100H (as this is not how the experiments are referenced in the main text). Figure 4 – Higher contrast between sediment and water would make this easier to differentiate. Here different run times are referenced which appear nowhere in the text. Figure 5 – H and L could be expanded to "high supply" and "low supply" . Consider rephrasing terms "normal" and "not normal". Provide sample sizes for each box plot and include labels for mean and standard deviation. Would eliminate need for additional tables. Figure 6 – Add "Calculated sediment transport efficiency" to y-axis label. Provide sample sizes for each box plot and include labels for mean and standard deviation. Would eliminate need for additional tables.

Table Comments: Table 1 – See Figure 1 comment. Table 2 – See Figure 4 comment. Table 3 – A bit confusing, I would maybe separate the GSDs as done in other tables. Table 4 – This isn't discussed much in the text. I'm also a bit worried about averaging over different timescales here and also whether or not the average is the best metric if the experiment is still moving towards equilibrium when sediment begins to exit the flume. It would be useful to see how the sediment transport rates vary as a function of time since the experiment begins. See general comments regarding time to equilibrium. Table 5 – See Figure 5 comment. Table 6 - See Table 3 comment. Table 7 – Not sure this adds very much, as this comparison with Lisle is not a main part of the

discussion.

Line comments (Apologies for some differences in style that arise here): General: The term here-in is used a number of times, I'd suggest removing all appearances of it Abstract: P1 1 - consider revising to "sedimentary deposits" P1 2 - remove "shape" Introduction: P1 14 - remove "the"; consider rewording to remove "new stimuli" P1 16 – remove "procilivity for adjustment" P1 21 – replace "that results from" with "due to" P2 3 – Remove sentence starting with "accordingly" P2 13 – Consider revising "The superposition of change upon a pre-existing mass"; a bit awkward P2 14 – Consider changing "Four pairs of experiments" to "Two sets of four experimental runs" Methods: P2 23 – consider changing to "each experiment" P2 25 – relative used twice in this sentence, consider rephrasing P2 25-30 – consider adding numbers to the list. That said, I'm not sure the list adds much here. P3 4 – Add comma after "at the beginning of the experiment" P3 8 – Change "to be output from" to "to exit" P3 8-10 – Consider rephrasing, is a bit unclear P4 3 –A randomForests is not, as I'm aware, a standard way to extract this data, so some citations here providing details of the model/method would be useful. Results: P5 1-10: I think this entire section can be made more clear and that providing the measured values in the text will help make the results read more directly. P6 10 – I don't think the authors have enough data to argue for a threshold change in behavior here. This transition could very well be a continuum that the authors may just be unable to capture given the data they've collected. I would be cautious using threshold here. P7 4-5: Saying the "two systems behave more similarly" is quite vague. Again, I think actually including the measured values in the text here would better demonstrate the differences between the experiments. P8 5 – Remove "a number of key observations can be made regarding the distribution of transport efficiencies". Rephrase next sentence to "The distribution of calculated transport efficiencies for...". Again, values here would help. Another option for rephrasing would be "The mean transport efficiency for GSD1 is XX% lower than for GSD2..." Discussion: P9 9-10 – Consider changing "poorly sorted" and "narrowly graded" to "broadly" and "narrowly" graded to make comparison more straightforward. P9 Equation 3 – small d remains

undefined in the text. P10 Equation 4 – Ds50 remains underfined, consider rewriting all references to median grain size with the same convention (even if they differ in original references) P10 25-30 – I have a very hard time following this section. Please consider rewriting to make calculation more explicit. P11 9-10 – Reconsider using poorly and well-sorted here and instead use broad and narrow GSD Conclusion: P12 23-34 – Consider replacing GSD1 and GSD2 with "narrow" and "broad" GSDs P12 4 – Revise to remove "as you increase"; Missing period.

---

## Referee Comment (RC3) · Anonymous Referee #3 · 1 Jul 2019

The authors conduct a set of experiments to explore how the largest grains influence the form and evolution of an aggradational channel. They conclude that the largest grains exert substantial influence, with larger coarse grains resulting in deposits of higher longitudinal slope. I think this is a valuable experiment, and am convinced that this work represents a valuable next step in understanding the importance of the coarsest grains in river channel morphodynamics. However, I found the presentation of the work somewhat lacking. I encourage the authors to do additional data analysis and then re-write the manuscript to support the discussion with more specific results.

MAIN POINTS

1. The Discussion section is outsized relative to the Intro, Methods, and Results. It feels quite speculative in light of the sparse data presented in the Figures and Tables.

[Figure]

Specific notes provided below.

2. In the final paragraph of the Discussion the authors summarize their findings as "3 lines of evidence for GSD2 as less stable":

a. Lower slopes (very effectively demonstrated), prograde more quickly (I don't see this demonstrated anywhere, though it seems that the authors have the water surface profiles extracted with which to easily create plots to demonstrate this).

b. Grains were more equally mobile due to a lower maximum threshold stress. (I don't see threshold stress quantified anywhere here, and it seems to me that any discussion of equal mobility should be supported by some sort of grain size data).

c. Fewer, and less persistent bedforms (I don't see this demonstrated anywhere, though it seems that the imagery the authors collected should allow them to demonstrate this in a figure without too much trouble).

3. The authors should thoroughly proof-read their re-submission. The language was unnecessarily complicated in many places in the manuscript. For example: "raw values for which are shown in. . ."; "The difference varying alongside discharge. . ."; ". . .the superposition of change upon a pre-existing mass. . ."

LINE NOTES

Line 26) I'm not convinced that armor formation is inherently degradational. Couldn't an armor form through selective deposition of only the coarsest grains from the supply GSD?

Presentation of Lane balance) This feels like a bit of a straw man, especially given the great set of papers that have come out of Eaton's lab recently. I wonder if a stronger introduction for this manuscript could focus more thoughtfully on the existing questions

about the role of the largest grains, and how the impact of the largest grains has the potential to be very different in aggradational systems (this paper) when compared to degradational systems (e.g. the Mackenzie and Eaton papers). Line 14) This reviewer has not thought about transport efficiency in this framework, and would have benefited from a bit more context. Transport efficiency is $\eta$ (eta), yes? What are the units? How should I think about it? Line 24) This hypothesis is quite vague. "Different transport regimes"? I would have assumed that referred to bedload vs suspended... Line 29) "...the superposition of change upon a pre-existing mass..." I'm not sure what the authors mean here.

Line 5) Is "relative sediment storage efficiency" the same as "transport efficiency"? (General Methods) When did the experiments end? How long were the runs?

Line 3) Given this description of the slope calculation, I think it would be very helpful to add several panels to Figure 4 depicting the method of slope calculation for early/middle/late stage profile evolution, showing the points of max and min elevation selected and length over which the slope is calculated. Along these lines, is it possible that the slope in the experiment varied along the profile? Is the channel concave? Line 9) I suggest using a statistical test to compare between runs. An ANOVA, perhaps?

Line 4) One example of a sentence that could use re-phrasing: "For both grain sizes more material is stored at lower discharges correlating to the steeper angles of the deposits". I think the authors mean "both grain size distributions" and I believe a more meaningful way of describing the relationship between storage and slope would be, "...at lower discharges, resulting in steeper sloped deposits." Line 10) Change to "...increases the transport efficiency of..."

Discussion (General Comments)

The discussion of grain size sorting, armoring, and partial mobility ought to be supported by data on the bed surface grain size in the experiment, but none were presented. Is it possible that the coarsest grains were preferentially deposited along the upstream end of the experimental channel? Was the grain size distribution of the outflow material the same as the feed? These data seem to be essential information if the authors plan to provide a detailed discussion of the impact of the coarsest grains on armoring, size selective transport, etc. The discussion of bar forms is interesting, though it is unsupported by the results, as currently presented. A set of images, a few simple calculations (e.g. sinuosity), would go a long way.

FIGURES

Fig 1) What is the scale of the experimental setup? That is a great thing to put on a figure of this sort.

Fig 2) Is it possible that the x axis scales are offset between Figure 2a and 2b? How can 100% of the grains be finer than ∼6 mm, yet > 3% of the mass is ∼8mm?

Fig 3) This figure would benefit from annotations. I couldn't figure out what the roughness elements were until I watched one of the associated videos. A multi-panel figure would help here: Start of experiment, showing roughness elements, progradation of deposit wedge, etc.

Fig 4) Needs horizontal and vertical scales.

Fig 5) Caption is confusing. What is "normal" relative sediment concentration?

Fig 6) What are (a) and (b)?

---

## Author Comment (AC1) · 16 Jul 2019

The general comments from Reviewer 1 are presented below. We would like to thank the author of these comments for their substantial contributions to the reorganisation and restructuring of the paper. These comments were very useful in helping us improve the clarity of the message in this paper, and we believe will strengthen the arguments made within. First, general comments are presented and then followed by specific comments we feel appropriate to address at this time.

[Figure]

**General Comments**

Reviewer: This a personal opinion. The article's title could be changed to something more appropriate. When I received the article I thought that it was related to large chains in the sense of boulders or macro-roughness elements. Given that most steep channels do have boulders and other (actual) large grains, and those are neglected in this study, the title was misleading to me. Again, this is a personal opinion but please consider it if you think the same.

*Author: This is an issue that had not occurred to us, but is useful to bear in mind. We had not considered the confusion that might arise from the overlap between our work and that in steep, jam-structure dependent streams.*

Reviewer: The article structure does not convey the information in a fluid manner. The introduction has little information about aggrading systems and it seems to me that it gives more importance to degrading systems. Although, I understand that the idea was to make clear that we know more about degrading systems more information and references to what we really know about aggrading systems is required. There are virtually no references to any study that may have discussed aggrading systems.

*Author: This feedback is vital for us; clarifying the issue and making sure the justification is present in a positive sense (i.e., what we actually know) is paramount for this study. As a result, we are working on re-structuring the introduction in order for the information to flow more smoothly. In addition, and in response to one of the specific comments, more background information and references will be added to ensure that the justification is presented in a clear and defensible manner. Furthermore, changes to the other sections will be made so the overall coherency of the article is maintained.*

Reviewer: The article presents the study using Lane (1955) balance expression. Then, the assumptions of this expression are called into question and by doing so the hypothesis is formulated. The problem is that Lane 1955 did not consider a mixture of

sediment and therefore does not intended to explain the responses of different GSD, even when they have the same D50. Only later in the paper, in the discussion (page 10, line 4), this is explained. So, as a reader, I had problems trying to understand why this is not explained right away. The major concern about this is how the information flows in the article.

*Author: Following from the previous comment, the section from the discussion will be relocated to the introduction to provide a more immediate justification for the study. Lane (1955) will still be introduced, but only as a tool to demonstrate the issues with using Church (2006) and the role of grain size distributions in bed material. We believe that this comment makes the mission statement clearer.*

Reviewer: The hypothesis needs to be reformulated. I understand the idea of the study is to compare responses to different GSD and boundary conditions. This was well developed in the text. However, if I just take the hypothesis, it doesn't say that. "We hypothesise that, like degrading systems, the presence of the large grains will result in different transport regimes, as in MacKenzie and Eaton (2017), and thus different channel morphodynamics and depositional slope" It says that it is just the presence of large grains, what about boundary conditions? The article shows that is not just the presence of these large grains but discharge is a fundamental control.

*Author: We agree that the hypothesis does not fully present the same ideas that this study addresses. Therefore, we will reformulate the hypothesis to better represent the information conveyed in the updated introduction, where we shall try to impress the importance of large grains upon the reader. In addition, the role of boundary conditions will be explicitly included.*

Reviewer: A lot of information about bed structure, for example bars, is given by the end of the discussion. There is no data about this and only observations. This should be presented in a more formal way.

*Author: This is a very important point that we will be integrating into the paper's re-*

sults section. Whilst this data is qualitatively available in the video supplements, the addition of static images to this paper will greatly improve the present these ideas and observations through the discussion section.

**Specific Comments**

Specific comments were also provided, and we address the most relevant of the non-technical comments here.

Reviewer: 2) Abstract - line 4 - Is it correct to talk about "fan" if we are in a 1D system? The fan part is where the system spreads and here it does not occur.

*Author: This system was designed to simplify a three-dimensional fan into a single slice that represents the system slope, like the experimental design of Guerit et al., 2014. So whilst the system is not fan-like, we believe it represents the fundamental interaction between surface organisation and slope in a manner that approximates self-formed deposits such as fans.*

Reviewer: 3) Introduction - line 11 - There is one problem when we use the discharge as a variable to explain a certain response. If we double (or 3X, 4X, ...) the channel width while holding the discharge we may have different geomorphological responses. Therefore, is not actually the discharge, but some other characteristic (e.g., unit discharge) what is better for comparisons. This may be discussed somewhere.

*Author: We used discharge as our system defining metric because it was the boundary condition we had control over in this case. Actual channel width varied in a manner that was not constant along the length of the flume, therefore specific discharge was avoided. Similarly water depths were unknown, so shear stress would not have been a useful metric.*

Reviewer: 11) Methods - general - It would be really interesting to analyze the evolution

of the slope, that is, change of slope in time. I was wondering if the experiment came to a final equilibrium slope, or how do you decide to finish an experiment. Do we find the mean slope by the end of the experiment or by the middle of it. A simple plot would answer these interesting questions.

*Author: Equilibrium slope was never explicitly reached, where the output matches the input rate of sediment; this value was approached but not used as a criterion in other studies (e.g., Eaton and Church , 2004). Instead, our experimental limit was set by the volume of sediment we could supply. We believed that the morphodynamics were similar enough over the course of the experiment that it was not undergoing a substantial flux. We will expand more upon the nature of the evolution of the deposits and also add a figure that shows the temporal pattern of the slopes presented in this study, if necessary.*

Reviewer: 18) Discussion - Page 10 - Line 10 - It would be interesting to consider a little discussion about what may happen if we have the same D84 and different D50.

*Author: We believe including this point, in the manner of the thought experiments of Parker and Toro-Escobar (2002), will complement the discussion section greatly. However, we will endeavour to limit our speculation as to the effects that these changes would have.*

---

## Author Comment (AC2) · 16 Jul 2019

The general comments submitted by Referee 2 are addressed below. We would like to thank the reviewer for their comments, especially those regarding the methods, as we think they will greatly improve the paper and its arguments.

**General Comments**

Reviewer: Reorganization of introduction - I think the introduction reads fairly well, but that further motivation could be provided by discussing the predictions of Lane's balance at the beginning of the article. One could use the idea that Lane's balance

would predict the same slope for a give D50, regardless of the rest of the GSD as a null hypothesis, then reference the known importance of large grains in degrading systems and the lack of complementary work on aggrading systems in order to more directly motivate this work. I think this reorganization could help to streamline the logical progression of the manuscript.

*Author: This feedback agrees with those made by the first referee, with a reorganisation of the information displayed in the introduction necessary to improve the communication of this importance. As a result, the structure is being re-written to relocate Lane (1955) and introduce Church's (2006) conceptualisation of it. We also agree that this will improve the flow of logic within this article.*

Reviewer: (1) How where the discharges determined? Are they specified to span the range of partial transport to full bed mobilization? It would also be useful to provide the calculated/estimated shear (or Shields) stresses related to each of these discharges of both flows. I'm aware that this may require some assumptions in relation to the sidewall correction, but given that most of the literature on this topic is presented in terms of Shields stress, it would be useful to also provide this estimate, especially for the discussion of relative transport capacity.

*Author: Discharges were determined based on initial conditions used during trial experiments, as well as their ease of calibration in setting up these experiments. The discharges were not calculated to correspond to any given shear stress value; as the deposit slope was set by the sediment transport dynamics, we were unable to predict the corresponding slopes. Without controlling the slope of the deposit, as is traditionally done in such experiments, we could therefore not relate discharge to shear stress during experimental design.*

Reviewer: It took me until halfway through the results to recognize that the multiple measures of slope presented were from different time steps following the onset of sediment transport out of the flume. How long were the experiments run after this point and

how were the experiments determined to be over? Was an equilibrium slope/transport rate reached or were adjustments still occurring when the experiment ended? If equilibrium was reached, how was it determined?

*Author: In these experiments, equilibrium was not a concept explicitly used in the determination of any experimental condition as sediment output had to be dried and weighed, which was used to feedback this information during the experiment. Therefore, we are careful to avoid usage of the term "equilibrium" in discussion of the dynamics involved here. As a result, the length of each experiment was solely determined by the volume of input material available for sediment feed. In addition, during the experiments themselves we used the morphodynamics as in-situ justification; we believed that they had not substantially changed between when sediment was output and the end of the experiment. Therefore, we believed that the system behaviour was not in flux when the sediment supply ran out.*

Reviewer: (3) For the slope-derivation, I think more information should be provided regarding the randomForests model, how it works, and the degree of user-specification it requires. How many images are input in order to determine the slope? How are the sub-classes determined? Are there uncertainties associated with these slope measurements based on the method or number of sample images input? A citation here providing the relevant background information could also help. The authors later report the mean slope and standard deviation for each experiment, but it is unclear if this is from multiple time slices (if so, how many?), multiple locations in the flume, or related to some uncertainty in the slope estimation. Organization-wise, I don't necessarily think this needs its own section in the methods. Alternatively, I might suggest splitting the methods section into (1) Experimental set-up, (2) Measurements, and (3) Slope derivation.

*Author: We greatly appreciate this advice, as it is very helpful in improving the layout of the relevant sections, and will be used to clarify the methods employed within the paper for the reader.*
Reviewer: (4) I find GS1 and GS2 not to be very informative variable names. I would suggest changing them to GSnarrow and GSbroad or something more information so it is easier for the reader to keep track of throughout the paper. Even H and L are a bit confusing to keep track of, but less so.

*Author: We will take this into consideration when we are restructuring the paper for clarity.*

Reviewer: Organization of the results section - I found this section to be a bit muddy, with parts ' of the motivation, methods, and discussion being mixed in. While I am okay with some intermingling of these sections, in this cas, I found it to make this particular section a bit difficult to follow. Below I've made some suggestions to streamline this section. (1) Move Lane's balance discussion to introduction. See above. (2) Move sediment transport efficiency calculation to methods. I would suggest adding this following the slope derivation. If Lane's balance has already been presented in the introduction, it would naturally follow to calculate the sediment transport efficiency. Introduction of this calculation in the methods would allow the authors to more cleanly step through the results. Again, some information of the number of samples used to make these calculations would be helpful (table 5).

*Author: These comments support the changes we have already made and incorporated into the technically corrected version of this paper, agreeing with the suggestions already made by handling associate editor.*

Reviewer: (3) This is a style thing, but I would suggest avoiding things like "Panel A of Figure 3 shows.." and instead simply say "There is a significant difference between equilibrium slopes as a function of the supplied grain size distribution (Figure 3A)." I think this would help with readability.

*Author: We agree that the manner of text in this piece can be improved, and we will update the relevant sections with clearer language.*

Reviewer: (4) Much of the information in the tables is not fully presented in the paper. I would recommend more explicitly discussing these results in the main text. Lots of the results are presented in a fairly vague way (e.g. - ". . .both systems retaining a higher proportion of sediment" even though the authors have quantified these effects more directly. I would suggest rephrasing to provide these values directly in the text (e.g. – ". . . in response to a doubling of sediment supply, both systems retained a higher proportion of sediment, XX% for the narrow GSD and XX% for the broad GSD." This in-text quantification would also help to clarify the main differences between the experiments.

*Author: We will employ these changes in order to make explicit the differences between the two grain size distributions, introducing these into the text will make the points easier to get across.*

Reviewer: Argument for large grains – While I find the argument that the transition between partial transport and full mobilization of the GSD drives the observed differences in slopes observed in the experiments reasonable, I am not entirely convinced that the data presented really show this. I agree given the results that D50 is a poor metric for predicting behavior in aggradation systems, but I think more could be done to support the argument of the importance of large grains. Do the authors have any observations from the experiments to be show this? For example, was the sediment exiting the flume sieved to determine the GSD of the transported sediment compared to the supplied sediment? Can the photos/videos of the bed be used to determine if there is significant sorting that arises during the experiments that may support this idea? I imagine that the videos could be used to track the mobility (or immobility) of the largest grains (or the bed surface as a whole) in the flume to better evaluate this idea.

*Author: The addition of this kind of data will likely improve the strength of our arguments. The most demonstrative data for this would likely be images taken from the videos, which we will be integrating in the results discussion to make clearer to the reader the differences in bar dynamics.*

Reviewer: The portion of the discussion where shear stress calculations are made is quite confusing. It is unclear what inputs are being used and what information is being drawn from the calculation. Specifically, this sentence is quite unclear "Equation 4 produces a shear stress 44.4% greater for entrainment of the D84 than the entrainment of the median in GSD1 than in GSD2." I assume the authors are solving for $\tau_{ri}$ with reference to the D84 of both GSDs, but the reference stress value and the actual calculated values should be made explicit to better support this point. Additionally here, a comparison to the estimated shear (shields) stresses applied in the experiments (see previous comment) would help to bolster this point.

*Author: Thank you for this comment, we will update the wording to better clarify this section. We will also add a table to include these values for the reader's ease. In addition, more work by Alain Recking (Recking, A., 2013. Simple Method for Calculating Reach-Averaged Bed-Load Transport. Journal of Hydraulic Engineering 139, 70–75. https://doi.org/10.1061/(ASCE)HY.1943-7900.0000653) will be used to support this point in conjunction with the aforementioned shear stress calculation.*

Reviewer: Discussion of bar formation and effects – Currently, I think this point of the discussion appears as an afterthought. While I agree that this might not be the main result of the paper, the authors describe the differences in bar presence and morphology between GS1 and GS2 experiments in order to support their conclusions regarding the role of large grains. If this is a main point to bolster the argument related to the importance of large grains, mapping of these bar formations and quantifying their differences between runs should be included in the methods/results sections of the manuscript. This discussion would be better supported with photos or measurements in the text to more clearly illustrate the argument made

*Author: This point has been echoed in other comments, and is definitely an improvement to be made to the results section. We are constructing a segment on the bars observed within these experiments, as currently the only reference available is located in supplementary materials. To that end, photos of the bed surface and of the morpho-*

*dynamics will be included to display these processes and differences visually.*

---

## Author Comment (AC3) · 17 Jul 2019

The general discussion points from Reviewer 3 are presented below. The comments provided by the reviewer were helpful in the improvements we will make to this paper's clarity and organisation, and we would like to thank them for their time and effort. In addition, specific comments are presented afterwards.

**General Comments**

Reviewer: 1. The Discussion section is outsized relative to the Intro, Methods, and Results. It feels quite speculative in light of the sparse data presented in the Figures

and Tables. Specific notes provided below. The discussion of grain size sorting, armoring, and partial mobility ought to be supported by data on the bed surface grain size in the experiment, but none were presented. Is it possible that the coarsest grains were preferentially deposited along the upstream end of the experimental channel? Was the grain size distribution of the outflow material the same as the feed? These data seem to be essential information if the authors plan to provide a detailed discussion of the impact of the coarsest grains on armoring, size selective transport, etc. The discussion of bar forms is interesting, though it is unsupported by the results, as currently presented. A set of images, a few simple calculations (e.g. sinuosity), would go a long way.

*Author: We agree that the relative sizes of the sections needs addressing, therefore we will restructure the introduction with material from the discussion (as also recommended by other reviewers) to more logically and evenly distribute the information. Furthermore, photos will be added into the results section to bolster the evidence provided for our arguments in the paper, rather than present in the supplemental information provided alongside the paper.*

Reviewer: 2. In the final paragraph of the Discussion the authors summarize their findings as "3 lines of evidence for GSD2 as less stable": a. Lower slopes (very effectively demonstrated), prograde more quickly (I don't see this demonstrated anywhere, though it seems that the authors have the water surface profiles extracted with which to easily create plots to demonstrate this). b. Grains were more equally mobile due to a lower maximum threshold stress. (I don't see threshold stress quantified anywhere here, and it seems to me that any discussion of equal mobility should be supported by some sort of grain size data). c. Fewer, and less persistent bedforms (I don't see this demonstrated anywhere, though it seems that the imagery the authors collected should allow them to demonstrate this in a figure without too much trouble).

*Author: We agree that the current presentation of data in the article leaves a large degree of speculation in the discussion section, aligning with comments of the other*

*reviewers. Currently the supporting evidence lies in the supplementary material, or not alongside the paper. To address this, we will include more data in the body of the paper in order to support these statements (for a) and c)). The role of shear stress has also been brought up by other reviewers, and we will include more calculations in order to support the statements made (e.g., Alain Recking's transport model from 2013).*

Reviewer: 3. The authors should thoroughly proof-read their re-submission. The language was unnecessarily complicated in many places in the manuscript. For example: "raw values for which are shown in. . ."; "The difference varying alongside discharge. . ."; ". . .the superposition of change upon a pre-existing mass. . ."

*Author: This is very useful suggestion in order to clarify the message that this paper attempts to convey, and we will make sure to simplify the sentence structure where necessary.*

**Specific Comments**

Specific comments were also provided, and we address those of a non-technical nature here.

Reviewer: Page 1 Line 26) I'm not convinced that armor formation is inherently degradational. Couldn't an armor form through selective deposition of only the coarsest grains from the supply GSD?

*Author: We will correct this statement to represent static armour in the classical degradational sense, but mention mobile armour as separate given that it does form under feed scenarios and represents a more organisational and flow dependent structure.*

Reviewer: Page 2 Presentation of Lane balance) This feels like a bit of a straw man, especially given the great set of papers that have come out of Eaton's lab recently. I wonder if a stronger introduction for this manuscript could focus more thoughtfully on

the existing questions about the role of the largest grains, and how the impact of the largest grains has the potential to be very different in aggradational systems (this paper) when compared to degradational systems (e.g. the Mackenzie and Eaton papers).

*Author: This comment aligns with the views of the other reviewers and is greatly useful as feedback. We will use Lane as an introduction to Church (2006) and the importance of large grains as the guiding principles for our work, instead. We will also endeavour to make clearer the potential role in aggrading systems.*

Reviewer: Line 14) This reviewer has not thought about transport efficiency in this framework, and would have benefited from a bit more context. Transport efficiency is $\eta$ (eta), yes? What are the units? How should I think about it?

*Author: We will add more detail to our definition of transport efficiency, as well as clarity in the organisation of the paragraph.*

Reviewer: Page 5 Line 5) Is "relative sediment storage efficiency" the same as "transport efficiency"? (General Methods) When did the experiments end? How long were the runs?

*Author: Relative sediment storage efficiency is the proportion of sediment stored, whereas the data presented was in fact the amount transported. We will update both the text and accompanying table in order to remove this confusion. We will also add more information about the conduct of the experiments to methods section.*

---

## Author Response (AR1)

**Author's Response**

**William H. Booker and Brett C. Eaton**

This document contains the replies to all comments provided by the reviewers for this paper, as well as a marked up version of the document containing all changes made to the text and structure of the original paper. We would like to thank the reviewers for their time and comments, as we believe they have contributed greatly to improving the quality of this paper through the individual recommendations and wholescale changes to paper structure. The following sections are separated by reviewer, and then by comment type. Additionally, changes not explicitly recommended by reviewers have been made during the considerable restructuring phase that accompanied the rewriting of the results and discussion. Furthermore, corrections were made to values that have been identified as mis-calculated.

**1 Reviewer 1**

**1.1 General**

Reviewer: This a personal opinion. The article's title could be changed to something more appropriate. When I received the article I thought that it was related to large chains in the sense of boulders or macroroughness elements. Given that most steep channels do have boulders and other (actual) large grains, and those are neglected in this study, the title was misleading to me. Again, this is a personal opinion but please consider it if you think the same.

Authors: We agree that confusion may arise from the wording of this title, and have changed aggrading to self-forming to imply their mobility (rather than confusion as boulders).

Reviewer: The article structure does not convey the information in a fluid manner. The introduction has little information about aggrading systems and it seems to me that it gives more importance to degrading systems. Although, I understand that the idea was to make clear that we know more about degrading systems more information and references to what we really know about aggrading systems is required. There are virtually no references to any study that may have discussed aggrading systems.

Authors: We agree that the introduction lacked discussion of the very systems it set out to replicate, although the direct field comparison is somewhat limited. Therefore, we have included additional reference to studies of aggrading channels as well as restating the focus of this paper. We have also re-structured the introduction in order for the information to flow more smoothly. In addition, and in response to one of the specific comments, more background information and references were added to ensure that the justification is presented in a clear and defensible manner. Furthermore, changes to the other sections were made so that the overall coherency of the article is maintained.

Reviewer: The article presents the study using Lane (1955) balance expression. Then, the assumptions of this expression are called into question and by doing so the hypothesis is formulated. The problem is that Lane 1955 did not consider a mixture of sediment and therefore does not intended to explain the responses of different GSD, even when they have the same D50. Only later in the paper, in the discussion (page 10, line 4), this is explained. So, as a reader, I had problems trying to understand why this is not explained right away. The major concern about this is how the information flows in the article.

Authors: Following from the previous comment, the section from the discussion has been relocated to the introduction to provide a more immediate justification for the study. Lane (1955) is still used, but only as

a tool to demonstrate the issues with using Church (2006) and the role of grain size distributions in bed material. We believe that this reorganisation makes the mission statement clearer.

Reviewer: The hypothesis needs to be reformulated. I understand the idea of the study is to compare responses to different GSD and boundary conditions. This was welldeveloped in the text. However, if I just take the hypothesis, it doesn't say that. "We hypothesise that, like degrading systems, the presence of the large grains will result different transport regimes, as in MacKenzie and Eaton (2017), and thus different channel morphodynamics and depositional slope" It says that it is just the presence of large grains, what about boundary conditions? The article shows that is not just the presence of these large grains but discharge is a fundamental control.

Authors: We agree that the hypothesis does not fully present the same ideas that this study addresses. Therefore, we will reformulate the hypothesis to better represent the information conveyed in the updated introduction, where we tried to impress the importance of large grains upon the reader. In addition, the role of boundary conditions was explicitly included.

Reviewer: A lot of information about bed structure, for example bars, is given by theend of the discussion. There is no data about this and only observations. This should be presented in a more formal way.

Authors: A formalised representation of the data was added into the results section to demonstrate the ideas discussed later on. Instead of relying solely on the qualitative data in the supplemental videos, we have added Figures 8 and 9 of the bed and major process observed during the experiments.

**1.2** Line Comments**

Reviewer: 1) Abstract - line 2 - there is no need to say "shape", it is already included in the distribution.

Authors: We have included this change.

Reviewer: 2) Abstract - line 4 - Is it correct to talk about "fan" if we are in a 1D system? The fan part is where the system spreads and here it does not occur.

Authors: This system was designed to simplify a three-dimensional fan into a single slice that represents the system slope, like the experimental design of Guerit et al., 2014. So, whilst the system is not fan-like, we believe it represents the fundamental interaction between surface organisation and slope in a manner that approximates self-formed deposits such as fans.

Reviewer: 3) Introduction - line 11 - There is one problem when we use the discharge as a variable to explain a certain response. If we double (or 3X, 4X, ...) the channel width while holding the discharge we may have different geomorphological responses. Therefore, is not actually the discharge, but some other characteristic (e.g., unit discharge) what is better for comparisons. This may be discussed somewhere.

Authors: We used discharge as our system defining metric because it was the boundary condition we had control over in this case. Actual channel width varied in a manner that was not constant along the length of the flume, therefore specific discharge was avoided. Similarly water depths were unknown, so shear stress would not have been useful metric.

Reviewer: 4) Intro - line 11 - There are several other references to this statement (sed supply and discharge controls)

Authors: We have provided additional examples.

Reviewer: 5) Intro - Most of the Intro - Generally only one reference is given for a certain statement. More

references are required. For example, when talking about armour layers (line 24) only Andrews and Parker, 1987 is cited.

Authors: The number of references has been increased throughout the introduction.

Reviewer: 6) Intro - Page 2 - Line 8 - References are needed for this statement.

Authors: Removed as part of the reorganisation.

Reviewer: 7) Intro - Page 2 - Line 9 - Would it be better to start the discussion with something relatively newer than Lane (1955)? The experiments are really interesting, but starting the analysis based on this relatively old study (there is more information available related to stream power). It doesn't mean that this expression is not important, but, it does not fit what we know about sediment mixtures.

Authors: This has been changed so the introduction of Lane's equation is the manner in which we introduce Church's 2006 relation. As such, we restructured the overall introduction.

Reviewer: 8) Intro - Page 2 - Line 21 - The text is confusing. However what? Please notice that the idea does not flow starting with "however". There are some equations, definitions, and other text that makes this "however" confusing.

Authors: Removed, and reincorporated to the text during the reorganisation.

Reviewer: 9) Intro - Page 2 - Line 23 - This is critical, Lane never said that this works for a sediment mixture, as you mentioned in the discussion. Therefore, up to this line, calling into question the assumption is not valid. please try to find another way to present the hypothesis.

Authors: This has been integrated into the restructured introduction.

Reviewer: 10) Methods - general - This is the strong part of the article, it was really interesting.

Authors: Thank you.

Reviewer: 11) Methods - general - It would be really interesting to analyze the evolution of the slope, that is, change of slope in time. I was wondering if the experiment came to a final equilibrium slope, or how do you decide to finish an experiment. Do we find the mean slope by the end of the experiment or by the middle of it. A simple plot would answer these interesting questions.

Authors: Equilibrium slope was never explicitly reached, where the output matches theinput rate of sediment; this value was approached but not used as a criterion in other studies (e.g., Eaton and Church, 2004). Instead, our experimental limit was set by the volume of sediment we could supply. We believed that the morphodynamics were similar enough over the course of the experiment that it was not undergoing a substantial flux. We expanded upon the nature of the evolution of the deposits. We have attached a plot of the temporal trends of slope, however we feel that this is not relevant to our discussion as it stands. As we have focussed more generally on the experiments, the temporal evolution of slopes have been excluded from our discussion.

Reviewer: 12) Results - page 6 - line 8 - This statement is only true for 0.1 ml  $s^{-1}$ . Notice that in panel a) for 0.2 ml  $s^{-1}$ , there is no "strong distinction"

Authors: We have corrected this.

Reviewer: 13) Results - page 7 - line 3 - Notice that you need to reference Table 5 when you talk about the efficiency

Figure 1: Temporal relationships of slope values, separated into  $\text{GSD}_{broad}$  on the left, and  $\text{GSD}_{narrow}$  on the right. Frames are paired by the run conditions in the topright corner of the  $\text{GSD}_{narrow}$  plot.

Authors: This is a mistake held over from also talking about the output efficiency (Table 4) rather than considering it from a storage efficiency lens. The reference to which has been changed in the text and table.

Reviewer: 14) Results - General - It would be good to have more information about the properties of the bars that are mentioned at the end of the discussion.

Authors: This section was added.

Reviewer: 15) Discussion - General - Some parts can be moved to the intro for a better motivation for the study. Also, it would help understand the hypothesis

Authors: The more introduction leaning sections, that improve message clarity, were moved to the beginning of the paper.

Reviewer: 16) Discussion - General - Like in the intro, more references are needed. It is generally poor in important references

Authors: More references have been included during the rewrite of the discussion section.

17) Discussion - General - I'm not making a lot of detailed comments in the discussion because it seems to me that in he new version it will change significantly. Only the most important specific points are considered here

Authors: The overall structure has changed, with the reorganisation encompassing other sections also.

Reviewer: 18) Discussion - Page 9 - Line 30 - It seems that Church's relation can be a better way to motivate the study.

Authors: agreed, and the introduction has been changed to reflect this.

Reviewer: 19) Discussion - Page 10 - Line 10 - It would be interesting to consider a little discussion about what may happen if we have the same D84 and different D50.

Authors: We have included a short discussion of the possibility of pairing experiments by a coarser grain, towards the end of the discussion.

Reviewer: 20) Discussion - Page 10 - Lines 12 to 17 - These lines are confusing. First, you mentioned that in low slopes sand plays an important role and that you can make the same inference. Then you said it is not actually sand what is the control in your experiments but the absence of large grains. Notice that your statement is correct (it is the absence of large grains), but relating it to Curran and Wilcock does not make any sense, because they attributed to sand

Authors: We clarified this, as the analogy is meant as a natural opposition to the mobility changes observed by Curran and Wilcock (2005). Instead, it is supposed to evoke the stabilisation effects of a mobile armour, or coarse bed organisation.

Reviewer: 21) Discussion - Page 10 -Lines 20 to 32 - A lot of confusing statements are given here. a) One important aspect that you are considering is channel slope. The analysis made using Eq. 4 does not include channel slope, even though it is known that slope plays a role critical shear stress Lamb has published a number of studies related to this topic. b) Comparing a change in critical stress change for D84 to a change in slope is confusing. Why can we do that? The problem is that for a given discharge if we vary slope water depth changes as well, therefore changes in slope and not directly comparable to changes in shear stresses. Maybe I'm missing something but if you explain a little more about this rationale it would be clearer.

Authors: a) We have clarified this position to introduce Recking (2013), which includes the role of slope on increasing Shields stress.

b) The original comparison was simply a neat similarity between the differences in the reference shear stress and slope, which when calculated using DuBoys' formula is directly related.

Reviewer: 22) Discussion - Everything related to bars and beyond reach average - Most of the text is not clearly related to data or measurements. It need to be better justified.

Authors: We have included a section of the results dedicated to this.

Reviewer: 23) Discussion - Page 12 - Line 6 - There are two more (more mobile)

Authors: We removed one 'more'.

Reviewer: 23) Conclusion - Page 12 - Line 34 - Change you in " as you increase". Also a period is missing.

Authors: We have included these changes.

**2 Reviewer 2**

**2.1 General**

Reviewer: Reorganization of introduction - I think the introduction reads fairly well, but that further motivation could be provided by discussing the predictions of Lane's balance at the beginning of the article. One could use the idea that Lane's balance would predict the same slope for a give D50, regardless of the rest of the GSD as a null hypothesis, then reference the known importance of large grains in degrading systems and the lack of complementary work on aggrading systems in order to more directly motivate this work. I think this reorganization could help to streamline the logical progression of the manuscript.

Authors: This feedback agrees with those made by the first referee, with a reorganisation of the information displayed in the introduction necessary to improve the communication of this importance. As a result, the structure was re-written to relocate Lane (1955) and introduce Church's (2006) conceptualisation as the crux of the argument. We also agree that this will improve the flow of logic within this article.

Reviewer: Methods clarification - While I generally follow the experimental set-up, I think some more detail can be provided regarding a few points. (1) How where the discharges determined? Are they specified to span the range of partial transport to full bed mobilization? It would also be useful to provide the calculated/estimated shear (or Shields) stresses related to each of these discharges of both flows. I'm aware that this may require some assumptions in relation to the sidewall correction, but given that most of the literature on this topic is presented in terms of Shields stress, it would be useful to also provide this estimate, especially for the discussion of relative transport capacity.

Authors: Discharges were determined based on initial conditions used during trial ex-periments, as well as their ease of calibration in setting up these experiments. The discharges were not calculated to correspond to any given shear stress value; as the deposit slope was set by the sediment transport dynamics, we were unable to predict corresponding slopes. Without controlling the slope of the deposit, as is traditionally done in such experiments, we could therefore not relate discharge to shear stress during experimental design.

Reviewer: (2) It took me until halfway through the results to recognize that the multiple measures of slope presented were from different time steps following the onset of sediment transport out of the flume. How long were the experiments run after this point and how were the experiments determined to be over? Was an equilibrium slope/transport rate reached or were adjustments still occurring when the experiment ended? If equilibrium was reached, how was it determined?

Authors: In these experiments, equilibrium was not a concept explicitly used in the determination of any experimental condition as sediment output had to be dried and weighed, in order to feedback this information during the experiment. Therefore, we are careful to avoid usage of the term "equilibrium" in discussion of the dynamics involved here. As a result, the length of each experiment was solely determined by the volume of input material available for sediment feed. In addition, during the experiments themselves we used the morphodynamics as in-situ justification; we believed that they had not substantially changed between when sediment was output and the end of the experiment. Therefore, we believed that the system behaviour was not in flux when the sediment supply ran out.

Reviewer: (3) For the slope-derivation, I think more information should be provided regarding the random-Forests model, how it works, and the degree of user-specification it requires. How many images are input in order to determine the slope? How are the sub-classes determined? Are there uncertainties associated with these slope measurements based on the method or number of sample images input? A citation here providing the relevant background information could also help. The authors later report the mean slope and standard deviation for each experiment, but it is unclear if this is from multiple time slices (if so, how many?), multiple locations in the flume, or related to some uncertainty in the slope estimation. Organization-wise, I don't necessarily think this needs its own section in the methods. Alternatively, I might suggest splitting the methods section into (1) Experimental set-up, (2) Measurements, and (3) Slope derivation.

Authors: We clarified the methods used in the derivation of the water surface slope, expanding upon the use of randomForests models which are typically used in more involved machine automation processes than simple supervised RGB classification. We have also updated the slope values with the number of observations used in each calculation (Table 1).

Reviewer: (4) I find GS1 and GS2 not to be very informative variable names. I would suggest changing them to GSnarrow and GSbroad or something more information so it is easier for the reader to keep track of throughout the paper. Even H and L are a bit confusing to keep track of, but less so.

Authors: We have updated the names of the mixtures, but the manner in which the experiments are referred is simple enough in our view.

Reviewer: Organization of the results section - I found this section to be a bit muddy, with parts of the motivation, methods, and discussion being mixed in. While I am okay with some intermingling of these sections, in this cas, I found it to make this particular section a bit difficult to follow. Below I've made some suggestions to streamline this section. (1) Move Lane's balance discussion to introduction. See above. (2) Move sediment transport efficiency calculation to methods. I would suggest adding this following the slope derivation. If Lane's balance has already been presented in the introduction, it would naturally follow to calculate the sediment transport efficiency. Introduction of this calculation in the methods would allow the authors to more cleanly step through the results. Again, some information of the number of samples used to make these calculations would be helpful (table 5).

Authors: The paper has been reorganised to result in a better flow of information throughout the paper. Primarily, this is through the relocation of the Lane and Church equations to a more justifying position at the beginning of the introduction. The number of observations are the same for all calculations using the slope (given in Table 1).

Reviewer: (3) This is a style thing, but I would suggest avoiding things like "Panel A of Figure 3 shows.." and instead simply say "There is a significant difference between equilibrium slopes as a function of the supplied grain size distribution (Figure 3A)." I think this would help with readability.

Authors: We have changed the occurences of this to a more passive presentation, that helps with the flow of information.

Reviewer: (4) Much of the information in the tables is not fully presented in the paper. I would recommend

more explicitly discussing these results in the main text. Lots of the results are presented in a fairly vague way (e.g. - ". . .both systems retaining a higher proportion of sediment" even though the authors have quantified these effects more directly. I would suggest rephrasing to provi

---

## Author Response (AR2)

**Author's Response**

**William H. Booker and Brett C. Eaton**

This document contains the replies to all comments provided by the reviewers for this paper, as well as a marked up version of the document containing all changes made to the text and structure of the last copy of the manuscript. We would like to thank the two reviewers for their comments, both for their continued help through this process and coming into this at the second stage. Both reviewers provided useful feedback not only for this current article but also more widely considering our research. The following sections are separated by reviewer, and then by comment type. Small revisions and corrections were also made where incorrect spelling or syntaxes existed.

**1 Reviewer 1**

**1.1 General comments**

Reviewer: I reviewed the first version of this article. This new version contains the core of all my suggestions and observations, I really appreciate the effort. It is also a much better article than the previous one. After careful consideration, I believe that it is worth publishing. I suggested technical corrections based on a series of typos, and some missed references that may improve the final article. It is a quite interesting topic and I enjoyed reading it.

*Authors: Thank you for your time and comments, we greatly appreciate both and, again, feel that your contributions have greatly improved our article.*

**1.2 Specific comments**

Reviewer: Page 2, line 16: There are several studies about organisation into cells. Please consider adding the following:
Dietrich, W. E., Kirchner, J. W., Ikeda, H., & Iseya, F. (1989). Sediment supply and the development of the coarse surface layer in gravel-bedded rivers. Nature, 340, 215217. `https://doi.org/10.1038/340215a0`
Dietrich, W. E., Nelson, P. A., Yager, E., Venditti, J. G., Lamb, M. P., & Collins, L. (2005). Sediment patches, sediment supply, and channel morphology. In G. Parker & M. H. Garcia (Eds.), 4th IAHR Symposium on River Coastal and Estuarine Morphodynamics RCEM 2005 (pp. 7990). Urbana, Illinois, USA: Taylor & Francis Group. `https://doi.org/10.1201/9781439833896.ch11`
Monsalve, A., & Yager, E. M. (2017). Bed Surface Adjustments to Spatially Variable Flow in Low Relative Submergence Regimes. Water Resources Research, 53(11), 93509367. `https://doi.org/10.1002/2017WR020845`
Nelson, P. A., Venditti, J. G., Dietrich, W. E., Kirchner, J. W., Ikeda, H., Iseya, F., & Sklar, L. S. (2009). Response of bed surface patchiness to reductions in sediment supply. Journal of Geophysical Research: Earth Surface, 114(2), F02005. `https://doi.org/10.1029/2008JF001144`
Nelson, P. A., Dietrich, W. E., & Venditti, J. G. (2010). Bed topography and the development of forced bed surface patches. Journal of Geophysical Research: Earth Surface, 115, F04024. `https://doi.org/10.1029/2010JF001747`

*Authors: We added some of these papers into the text, in support of the statements concerning cells and patches.*

Reviewer: Page 2, line 20: Please consider the following references

Dietrich, W. E., Kirchner, J. W., Ikeda, H., & Iseya, F. (1989). Sediment supply and the development of the coarse surface layer in gravel-bedded rivers. Nature, 340, 215217. `https://doi.org/10.1038/340215a0`

Nelson, P. A., Venditti, J. G., Dietrich, W. E., Kirchner, J. W., Ikeda, H., Iseya, F., & Sklar, L. S. (2009). Response of bed surface patchiness to reductions in sediment supply. Journal of Geophysical Research: Earth Surface, 114(2), F02005. `https://doi.org/10.1029/2008JF001144`

Nelson, P. A., Dietrich, W. E., & Venditti, J. G. (2010). Bed topography and the development of forced bed surface patches. Journal of Geophysical Research: Earth Surface, 115, F04024. `https://doi.org/10.1029/2010JF001747`

*Authors: We added some of these references.*

Reviewer: Page 3, line 11. Parenthesis was misplaced. It shoud say ... within a mixture (MacKenzie et al ...)

*Authors: We have corrected the text.*

Reviewer: Page 3, line 13. It should say "is to test"

*Authors: We have corrected the text.*

Reviewer: Page 8, first line. References to Table 1 and 2 are misplaced. It should say Mean slopes are given in Table 1 .... size distribution are shown in Table 2.

*Authors: We have reordered the table references.*

Reviewer: Page 8, line 6. Typo It should say 59.3%

*Authors: We have corrected this in text.*

Reviewer: Page 8, line 10. Table 5 is used before Table 4.

*Authors: We have corrected the ordering of the tables.*

Reviewer: Page 9, line 2. You need to say something like "For GSDbroad the D50..." Notice that you are later comparing it to GSDNarrow.

*Authors: We have included this suggestion for clarity.*

Reviewer: Page 11, line 24. "bed material, and load, exerts..." Do you need the first comma ?

*Authors: We removed the clause as it was unnecessary.*

Reviewer: Page 15, line 19. There is a strange sentence, currently it says "We see the output Our calculations..." It makes no sense.

*Authors: We removed the "We see the output"; it was an incomplete sentence that was not removed before compilation.*

Reviewer: Page 26, Table 6. "n" is not defined in the text.

*Authors: We have added a definition to the table text.*

**2 Reviewer 2**

**2.1 General comments**

Reviewer: Overall, the authors have done a good job addressing the concerns of the three previous reviewers (of which I am not one) and therefore I recommend minor revisions and a slight restructuring of few paragraphs throughout the text. My opinion of the paper though is a bit more critical and I leave these more critical comments as something for the authors to consider as they move beyond the current manuscript.

*Authors: We would like to thank you for your time and considered comments, they have given us food for thought for both this paper and relevant experiments moving forwards.*

Reviewer: A major shortcoming of the manuscript is the reliance on discharge as the governing forcing variable with no measurement of the mobility thresholds of the two different grain size distributions through time. Measuring the system average shear stress and back calculating the threshold from flux measurements would seem to be a critical piece to understanding how variation in discharge relates to grain size mobility. At the very least assessing the particle mobility following the work of Wilcock and Colleagues via a mixed grain size transport equation would provide quantitative assessment of the output. It is not clear that slope really should be the primary variable of concern as the experiments show an adjustment of slope, flow depth, and surface GSD which would suggest that a Shields stress is a more apt term for comparison (transport capacity - Shields stress over the threshold - may be even more relevant). These shortcomings limit the transferability of the results here to other systems or experiments. I am not sure there is much that the authors can do about this so it is left as something to consider for future experiments.

*Authors: We agree that to cement the arguments in this paper, providing these data would be very useful. Unfortunately, we lack the data needed to make these calculations; if these experiments were to be performed again they would include the necessary modifications. The difficulty with this system, other than its operational constraints, is the presence of multiple bars which offer multiple sites of deposition and thus a difficulty establishing a relationship between the output material and channel morpodynamics given the extra, intermediate step. We do, however, argue that slope is still a useful indicator of the system, as we believe it provides the system scope determinants of energy that is then moderated by surficial adjustments. Although, direct measurements of shear stress and Shields stress would have provided substantially more information.*

Reviewer: A second potential shortcoming of the experiments is that they were not run to an equilibrium or steady state slope and sediment flux output. The cited models of Lane and Church (from my understanding) are for the final steady state profiles. If allowed to run to steady state the grain size distribution of the sediment bed for GSDbroad would need to coarsen and the slope would steepen in order for the flume system to conserve mass (the lack of bed surface GSD precludes measuring this). This is a standard feature of feed flumes (Parker and Wilcock, 1993; 1995). This equilibrium profile may reflect a more realistic grain size distribution for the standard surface based transport equations, which hinge on the transport capacity (shear stress or shields stress relative to the threshold). One way around some of these issues is the adherence to the experiments only exploring 'adjusting' or 'forming' reaches and not the final state, which I believe is their goal. However, this limits the experiments applicability when discussing concepts developed for or at steady state conditions.

*Authors: We agree that given enough time, the system might in fact change its bed surface in a substantial enough manner to result in changes to its character. However, whilst this may be a standard feature of flumes we remain unconvinced that this will necessarily occur in a system that has secondary modes of channel roughness adjustment, given their relative importance in controlling sediment transport. In particular this comment raised an interesting discussion on the relative timescales of adjustment, as led by the relative ability of the system to mobilise material towards its equilibrium state, and the difference between adjustment and equilibrium phases, for which we thank the reviewer for starting.*

Reviewer: Overall, the experiments are interesting and the exploration of the role of the GSD is a worthwhile

endeavor. The experimental results are not quantified in a way to close the loop as to what the variables are that one would need to measure in order to accurately predict average transport rates in these systems. Maybe it is the D84 as suggested, but that could be shown in experiments where these quantities can be measured. While the majority of my major comments are fairly critical, I recommend minor revisions as some of these things cannot be addressed with the current data and the authors have addressed the initial review and made substantial revisions. Within these minor revisions, I hope the authors consider the nuance of the models and frameworks that they are challenging and place their results into a context that adds to those models or paints the boundary conditions of those systems more systematically rather than dismissing them.

**2.2   Specific comments**

Ln. 5 - missing 'the'. '...that the largest'

*Authors: We have corrected the text.*

Reviewer: P. 2, Ln. 25 - It is not clear how the work of Parker (1990) and Wilcock and Crowe relate to channel stability. These works produce surface-based bed load transport relations, to my knowledge these equations have not been connected to a channel with deformable banks (i.e. not a flume). Maybe bed stability is the more apt concept here?
For channel stability these works would only likely be valid through the work of Parker (1978, 1979) and that would require the bed to be run to equilibrium under sediment feed conditions at which point (like a fixed wall flume) the coarse grains (if sufficiently less mobile to begin with) could accumulate to build a channel in which they can be transported and changing the grain size distribution. Otherwise it is hard to understand how mass would balance.

*Authors: We agree this distinction should be made explicit in text and have amended "channel" to "bed", both for clarity and to get closer to the mechanism proposed by MacKenzie and Eaton (2017).*

Reviewer: P. 3, Ln. 7-20. Could you provide an analogue for what you mean by an aggrading system? Fans are mentioned, but it is not clear how this experimental set up recreates those conditions. In the experiments by Delorme et al. (2017) for a bimodal system two fans were created, a coarse one and a fine one which is similar to many fan profiles. Why wouldn't your system simple recreate this phenomena given sufficient time and space? Is this just an aggrading reach then, because there are no downstream effects which are important for fans. The experiments of Guerit et al stop when they reach the outlet, because deposition is a key feature of fans and at the outlet their experiment becomes a steady state profile.

*Authors: We added a sentence into the introduction more generally definins aggrading systems. As an anecdotal natural analogue to this setup, the current channel in Cougar Creek in Canmore, AB shows a similar planform of bars between engineered banks. During floods the material is highly mobile and otherwise remains unvegetated, bearing a striking resemblance to the morphology exhibited here. The system here could either be directly compared to a confined, aggrading reach or a one (and a half) dimensional representation of a fan channel, in much the same way a 1D hydraulic model represents a river channel. More broadly, however, this setup represents fans through a profile approach, where the loss of competence through flow expansion is mimicked by the change in slope caused by the foam insert. Thus holistic, system scale controls over the behaviour and form of the deposit are likely to represent system scale differences between boundary conditions. It is unclear, during the running of these experiments, whether downstream fining would occur given the length of the flume, so given enough time in a larger system this may well come true. It was the morphodynamics of the experiment that prompted us to carry on with these experiments, given the scale of the flume we were not expecting the observed degree of difference between these experimental conditions. The behaviour of the deposit surface became the focus of the study, although not initially intended, as it also represents differences in system not captured in the slope, as mentioned in your general comments.*

Reviewer: Due to the name change of the GSDs the video titles no longer match the manuscript. You might

just add a line in the manuscript that says (GSDnarrow = GSD2), and same for the broad one as well where the videos are first mentioned this way the videos online don't need to be changed. They are neat to watch.

*Authors: Thank you for bringing the latter to our attention too, we have added an addendum in the "Video Availability" section notifying the reader of this.*

Reviewer: P. 9, Ln. 1. feed load instead of fed load? Also in line 6, missing an 'L' in overall.

*Authors: We changed the use of "fed" to input, for clarity, and corrected the misspelling.*

Reviewer: Equations 4 & 5. Could you report the actual values that were calculated rather than just the percent differences? Or put them in one of the tables? One set of these is in a table, but it would be worthwhile to report them in the text as well.

*Authors: Equation 4 only returns a ratio of grain sizes that are reported in Fig. 2, here given as percent values in text, and we have added the numeric values of Eq. 5 to the text.*

Reviewer: P. 13, Ln. 5-10. This is a straw man argument, that doesn't benefit the paper's key results. Equal mobility is only valid in some cases, the many papers of Wilcock and colleagues (cited within the manuscript) demonstrate partial mobility and fractional transport are likely the norm at low shear stresses. Additionally, the work of Paola and Seal (1995) on particle sorting demonstrate that the tails of the reach GSD can't follow equal mobility in order for sorting profiles to occur (even under an aggrading system sorting can still occur following the experiments of Delorme et al. 2017 as an example)

*Authors: This argument is established in so much as hiding is explicitly the function mentioned in Church (2006) as the main theoretical driver for equalising entraining stresses and thus the justification for his use of the $D_{50}$, and ultimately why we observe the differences in slope between the two GSDs. However, we agree that it does not substantively add to the discussion following after and so we chose to remove this paragraph.*

Reviewer: P. 14, Ln. 6-8. This statement requires evidence or citations, I only say this because the data exist to prove this so the speculation here isn't particularly useful. As far as I have seen, fans tend to have strong sorting profiles such that the upper reaches are coarse and become finer downstream.

*Authors: We have altered the wording of this passage to reflect the role of the experimental design and observed outcomes in other experiments.*

Reviewer: P. 14, Ln. 10. In my opinion this is not a particularly nuanced discussion of the models proposed by Lane or Church, which from my reading are implicitly for the equilibrium profile following the adjustment phase. The current experimental systems are stopped when the system runs out of sediment as determined by availability constraints (?) (is this just a logistical issue?). At equilibrium the model may actually match much closer to that of church, b/c at equilibrium or steady state the slope will reflect that needed to transport the coarse grains and the bed will have to reflect a coarser size distribution. This is a basic feature of feed flume systems (see Parker and Wilcock, 1993; 1995)

*Authors: This point has proved very useful in changing our understanding of the outcomes of these experiments, and resulted in several changes throughout the paper. We agree that continued experimental run time would clarify the issue of bed state influence, as the graded state of Lane (and thus Church) can only be considered over long periods of time. We operated under the assumption, however, that the graded state would be more closely approximated by the initial depositional state of these experiments because they were allowed to set their own slope, under the given boundary conditions and so differences in character would be demonstrated. Your latter point especially bears consideration, as flume studies would suggest surface responses to counteract the output changes. However, as the system has the addition of a secondary method by which sediment transport can be mediated (lateral segregation) that represents a magnitude of bed mobility*

*changes on the order of, if not larger, than the surface adjustments only available to a flume. The interaction between the grain and bed form scale modes of stabilisation are something of great interest, and we believe should be studied further in order to elucidate their relative importances.*

Reviewer: P. 16, Ln. 8. I am having a hard time following the the channel geometry arguments as there is no analysis of channel geometry within the manuscript, nor are their banks within the flume which seem like a prerequisite feature for understanding channel geometry.

*Authors: The channel geometry argument of Pfeiffer et al. (2017) was invoked in order to explain the differences in observed slopes, in an attempt to link these experiments more holistically to competence and capacity driven deposition. However, the link is difficult to justify and not especially well made, so we chose to remove this paragraph from the article.*

Reviewer: P. 16, Ln. 16. Coarse-grained natural systems seem to be limited to a range of 1 to 7 in terms of the shear stress relative to the reference threshold shear stress (Phillips & Jerolmack, 2016). This would support the lower discharge claim, but it is also not clear what the actual shear stress range relative to be mobility is within this system.

*Authors: This is an interesting point, and situates this work within a wider context. As mentioned, this point is difficult to demonstrate, however, given our lack of shear stress estimations.*

Reviewer: P. 16, Ln. 24. The entire flume width is not submerged, this is the expected behavior in a flume or a very confined stream table type of experiment which this may be more analogous to.

*Authors: We have adjusted the text to demonstrate the expectation and included an alternate example.*

[revised manuscript text omitted]
 | 1.87 x 10$^{-2}$ | 4.42 x 10$^{-4}$ | 224 | 4.75 x 10$^{-2}$ | 2.21 x 10$^{-3}$ | 281 |
| 100H | 3.21 x 10$^{-2}$ | 2.81 x 10$^{-3}$ | 101 | 4.05 x 10$^{-2}$ | 3.82 x 10$^{-3}$ | 175 |
| 200L | 7.71 x 10$^{-3}$ | 2.48 x 10$^{-4}$ | 437 | 6.60 x 10$^{-2}$ | 3.29 x 10$^{-3}$ | 449 |
| 200H | 6.33 x 10$^{-2}$ | 5.50 x 10$^{-3}$ | 256 | 7.68 x 10$^{-2}$ | 2.80 x 10$^{-3}$ | 243 |

**Table 7.** Changes in mean transport efficiency for the column name experiment, given relative to the row name experiment, for GSD$_{broad}$ and GSD$_{narrow}$ respectively. Values are given in percent.

|      | 100L | 100H | 200L | 200H |
|------|------|------|------|------|
| 100L | - | -58.7/39.1 | 71.9/-14.7 | 239/61.9 |
| 100H | 142/-28.1 | - | 316/-38.6 | 721/16.4 |
| 200L | -41.8/17.2 | -76.0/63.0 | - | 97.3/89.8 |
| 200H | -70.5/-38.2 | -87.8/-14.1 | -49.3/-47.3 | - |

**Table 8.** Mixture mobility transition points calculated using Eq. (5), taken from Recking (2013).

|      | $GSD_{broad}$ | $GSD_{narrow}$ |
|------|---------------|----------------|
| 100L | 0.414         | 0.318          |
| 100H | 0.479         | 0.417          |
| 200L | 0.228         | 0.240          |
| 200H | 0.225         | 0.264          |

**Table 9.** Flume dimensions and run conditions for Lisle et al. (1991) and the two 100L experiments included here.

|                        | Lisle et al. (1991) | $GSD_{broad}$ | $GSD_{narrow}$ |
|------------------------|---------------------|---------------|----------------|
| Length (m)             | 7.5                 | 2             | 2              |
| Width (m)              | 0.3                 | 0.128         | 0.128          |
| Slope (m/m)            | 0.03                | 0.083         | 0.058          |
| Grain Size Range (mm)  | 0.35-8              | 0.25-8.0      | 1.4-2.8        |
| $D_{50}$ (mm)          | 1.4                 | 2.03          | 2.02           |
| Flow Rate (ml s$^{-1}$)| 582                 | 100           | 100            |
| Feed Rate (g s$^{-1}$) | 8.4                 | 1             | 1              |
| Run Time (min)         | 560                 | 549           | 429            |

---

## Author Response (AR3)

**Author's Response**

William H. Booker and Brett C. Eaton

This document contains our reply and acknowledgement of the technical corrections requested by the associate editor Jens Turowski, included in the submitted manuscript. We would like to thank Jens for his corrections and help throughout the process. In addition, the ackowledgements section has been populated and the changes to figure names in the text have been reflected in the Code and Data Availability section.

**Technical Corrections**

11.30 The extent varies

*Authors: We have made the correction.*

13.6 the bed states that

*Authors: We have corrected the text*

15.6 mobility

*Authors: We have corrected the spelling.*

[revised manuscript text omitted]